# Genetically controlled mtDNA deletions prevent ROS damage by arresting oxidative phosphorylation

Simon Stenberg[1,2], Jing Li[3,4], Arne B Gjuvsland[1], Karl Persson[2], Erik Demitz-Helin[2], Carles González Peña[2], Jia-Xing Yue[3,4], Ciaran Gilchrist[2], Timmy Ärengård[2], Payam Ghiaci[2], Lisa Larsson-Berglund[2], Martin Zackrisson[2], Silvana Smits[2], Johan Hallin[2], Johanna L Höög[2], Mikael Molin[2,5], Gianni Liti[4], Stig W Omholt[6]*, Jonas Warringer[2]*

[1]Centre for Integrative Genetics, Department of Animal and Aquacultural Sciences, Norwegian University of Life Sciences, Ås, Norway; [2]Department of Chemistry and Molecular Biology, University of Gothenburg, Gothenburg, Sweden; [3]State Key Laboratory of Oncology in South China, Collaborative Innovation Center for Cancer Medicine, Guangdong Key Laboratory of Nasopharyngeal Carcinoma Diagnosis and Therapy, Sun Yat-sen University Cancer Center, Guangzhou, China; [4]Université Côte d'Azur, CNRS, INSERM, IRCAN, Nice, France; [5]Department of Biology and Biological Engineering, Chalmers University of Technology, Gothenburg, Sweden; [6]Department of Circulation and Medical Imaging, Cardiac Exercise Research Group, Norwegian University of Science and Technology, Trondheim, Norway

*For correspondence:
Stig.omholt@ntnu.no (SWO);
jonas.warringer@cmb.gu.se (JW)

Competing interest: The authors declare that no competing interests exist.

**Abstract:** Deletion of mitochondrial DNA in eukaryotes is currently attributed to rare accidental events associated with mitochondrial replication or repair of double-strand breaks. We report the discovery that yeast cells arrest harmful intramitochondrial superoxide production by shutting down respiration through genetically controlled deletion of mitochondrial oxidative phosphorylation genes. We show that this process critically involves the antioxidant enzyme superoxide dismutase 2 and two-way mitochondrial-nuclear communication through Rtg2 and Rtg3. While mitochondrial DNA homeostasis is rapidly restored after cessation of a short-term superoxide stress, long-term stress causes maladaptive persistence of the deletion process, leading to complete annihilation of the cellular pool of intact mitochondrial genomes and irrevocable loss of respiratory ability. This shows that oxidative stress-induced mitochondrial impairment may be under strict regulatory control. If the results extend to human cells, the results may prove to be of etiological as well as therapeutic importance with regard to age-related mitochondrial impairment and disease.

## Editor's evaluation

Stenberg et al., explore how cells adapt to mitochondrial oxidative stress using yeast as their model system. The authors propose that reversible loss of mtDNA leads to reduced ETC function and diminished free radical production and that this represents an evolved survival mechanism. The idea that reversible loss of mtDNA may be an adaptive response under genetic control that is triggered to permit survival under adverse conditions where oxidative stress is elevated is novel and potentially important, especially given critical questions on how chronic or acute oxidative stress may contributes to loss of mtDNA integrity and mitochondrial dysfunction.

## Introduction

Mitochondrial impairment is strongly associated with aging (*Sun et al., 2016*) and the pathogenesis of age-related human diseases, including Alzheimer's disease (*Hu et al., 2017*), Parkinson's disease (*Ammal Kaidery and Thomas, 2018*), the deterioration of skeletal and cardiac muscle (*Hepple, 2016*), and macular degeneration (*Hyttinen et al., 2018*). Mitochondrial DNA (mtDNA) deletions are perceived to contribute markedly to this impairment (*Krishnan et al., 2008*). The general conception is that mtDNA deletions are deleterious events related to either faulty mtDNA replication or the mis-repair of mtDNA of double-strand breaks (*Nissanka et al., 2019*). However, considering that mtDNA deletions are prone to cripple oxidative phosphorylation (*Fontana and Gahlon, 2020*) (OXPHOS), and thus turn off the production of intramitochondrial superoxide anion ($O_2^{\cdot-}$), it is conceivable that mtDNA deletion is also under genetic control. The main rationale is that compared to a mitophagic response (*Bess et al., 2012*; *Palikaras and Tavernarakis, 2014*; *Sedlackova and Korolchuk, 2019*; *Gustafsson and Dorn, 2019*; *Ng et al., 2021*), it would serve as a less costly defense mechanism against an abrupt increase in reactive oxygen species (ROS) inundating the primary antioxidant defenses (*Sies et al., 2017*; *Shpilka and Haynes, 2018*). The disclosure of an additional genetically controlled defense layer against $O_2^{\cdot-}$ damage, situated between the primary antioxidant defenses and mitophagy, would bring a fresh perspective to what causes mitochondrial impairment and how it can be mitigated. This motivated us to search for the existence of such a regulatory layer in wild budding yeast (*Saccharomyces cerevisiae*).

## Results

### Paraquat impairs cell growth through mitochondrial $O_2^{\cdot-}$ production

Domestication has systematically enhanced fermentative and reduced respiratory asexual growth (*De Chiara et al., 2020*), with common lab strains harboring multiple defects in mitochondrial respiratory biology (*Gaisne et al., 1999*). To avoid possible confounding effects of domestication we therefore chose to work with the wild strain YPS128 (*Liti et al., 2009*). We exposed haploid YPS128 cell populations expanding clonally on glucose to the mitochondrial $O_2^{\cdot-}$ generator and redox cycler paraquat (N,N-dimethyl-4–4'-bipiridinium dichloride) (*Cochemé and Murphy, 2008*). Paraquat generates $O_2^{\cdot-}$ by passing electrons from OXPHOS complex III and mitochondrial NADPH dehydrogenases to $O_2$ (*Cochemé and Murphy, 2008*; *Castello et al., 2007*), a mode of $O_2^{\cdot-}$ generation that is a good proxy for the in vivo situation (*Zou et al., 2017*).

We titrated the paraquat dose to cause a 2.5-3 fold increase in cell doubling time (*Figure 1—figure supplement 1A*). At the chosen concentration (400 µg/mL), key mitochondrial oxidative stress response genes, copper/zinc dependent $O_2^{\cdot-}$ dismutase (Cu/ZnSOD, Sod1), manganese-dependent $O_2^{\cdot-}$ dismutase (MnSOD, Sod2) and mitochondrial cytochrome C peroxidase (Ccp1), increased their transcript levels 2–12-fold in early lag-phase (*Figure 1—figure supplement 1B*). The cells maintained the elevated expression of these antioxidant transcripts throughout the exponential growth phase and the following two growth cycles without causing any detectable reduction in the cell doubling time. Assuming that this expression increase reflects mobilization of the whole repertoire of primary antioxidant defenses, the lack of growth improvement shows that the paraquat-induced $O_2^{\cdot-}$ production was well above the reach of these defenses, while still allowing for cellular function.

Addition of vitamin C, an antioxidant that accepts electrons from PQ⁺, the free radical (or 'damaging') state of paraquat (*Sendra et al., 1999*), caused the growth of paraquat exposed wild type cells, as well as paraquat exposed *sod2Δ* cells, to be on par with that of unexposed wild type cells (*Figure 1—figure supplement 1C, D*). This suggested that a paraquat concentration of 400 µg/mL impaired cell growth entirely through its effect on $O_2^{\cdot-}$ production.

### Swift adaptation to increased $O_2^{\cdot-}$ production

We then used a high throughput growth platform (*Zackrisson et al., 2016*) to observe how 96 asexually reproducing yeast cell populations (colonies) on solid agar medium adapted to the chosen paraquat dose in terms of change in cell doubling time. The evolution experiment was run for 50 growth cycles, from lag to stationary phase (*Figure 1—figure supplement 1E*), with each cycle lasting 72 hr. The number of cells in each colony doubled 2.5–6 x in each cycle, and over the 50 growth cycles the populations doubled in size ~240 times, on average. Neglecting cell deaths and assuming synchronous cell

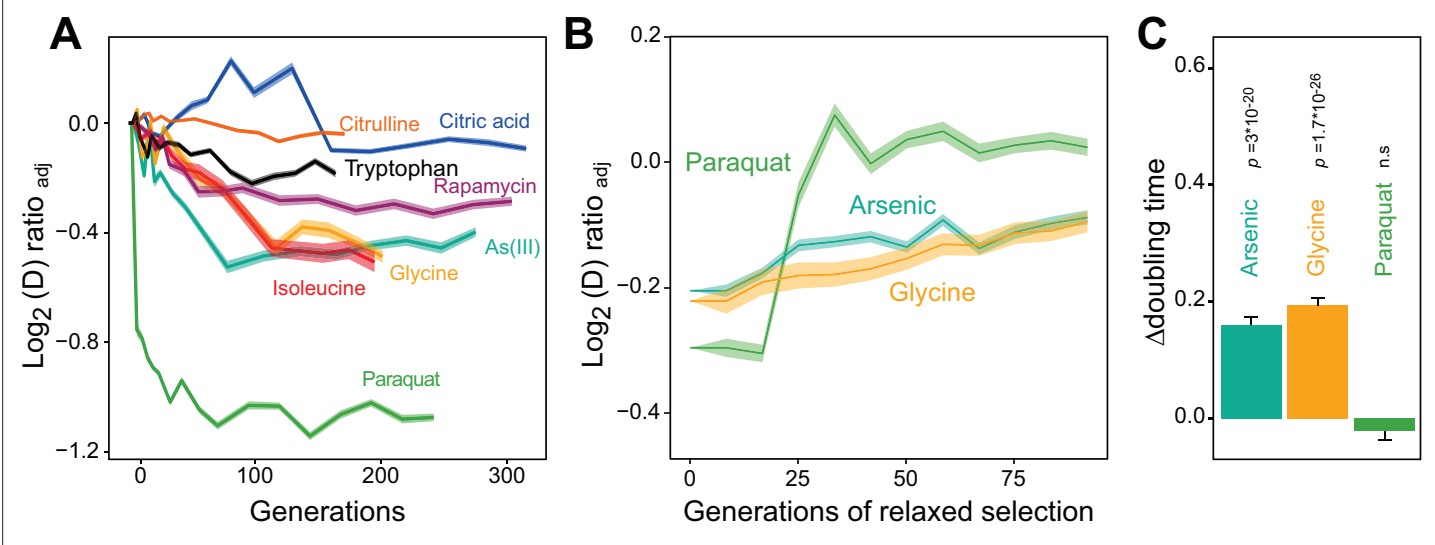

**Figure 1.** Distinct adaptation to paraquat. (**A**) Mean temporal adaptive response to paraquat and seven other stressors. y-axis shows $\log_2$ fold reduction in cell doubling time (**h**) from pre-stress, adjusting for plate, position and pre-culture effects. 96 populations for each stressor (n=6). Shade: S.E.M. (**B**) Loss of the acquired adaptation as a function of number of cell generations after release from the selection pressure. Colored lines: mean of 96 populations (each measured at n=5). Shade: S.E.M. The populations were released from stress after reaching 70–90% of their endpoint ($t_{50}$) adaptation. (**C**) The difference in cell doubling time (**h**) in a no-stress environment between 96 populations (each measured at n=5) having achieved 70–90% of their endpoint adaptation to paraquat, arsenic and glycine, respectively, and the founder population. The difference reflects the selective advantage of losing the acquired adaptations when the populations are no longer exposed to stress. p-values: one-sided t-test. Error bars: S.E.M. See also *Figure 1—figure supplements 1 and 2*.

The online version of this article includes the following source data and figure supplement(s) for figure 1:

**Source data 1.** Doubling time data of 96 populations adapted to each of eight different environments over *G* generations; doubling times are in the respective selection environment.

**Source data 2.** Difference in doubling time in absence of stress and in the respective selection environment, for adapted populations having achieved 70-90% of their final adaptation.

**Source data 3.** Doubling time data of 96 populations adapted to paraquat, arsenic and glycine over $G_s$ generations and then released from selection for $G_r$ generations; doubling times are in paraquat, arsenic, and glycine, respectively.

**Figure supplement 1.** Titration of the paraquat (PQ) dose and design of adaptation experiment.

**Figure supplement 1—source data 1.** FPKM data of selected oxidative defense genes, obtained from RNA-sequencing of cells exposed to paraquat.

**Figure supplement 1—source data 2.** Doubling time data of WT, *mip1Δ* and *sod2Δ* populations, with and without paraquat and with and without vitamin C.

**Figure supplement 1—source data 3.** Doubling time data of WT in different concentrations of paraquat.

**Figure supplement 2.** Comparison of predicted paraquat adaptation with the experimental data.

**Figure supplement 2—source data 1.** Doubling time data for the BY4741 single gene deletion collection under paraquat exposure; used as input for simulations in *Figure 1—figure supplement 2*.

**Figure supplement 2—source data 2.** Doubling time data of disomic strains growing in paraquat; used as input for simulations in *Figure 1—figure supplement 2*.

divisions, this corresponds to ~240 cell generations. To provide a comparative data set, we similarly exposed a total of 672 cell populations to seven other stressors not explicitly challenging mitochondrial function over 50 growth cycles (*Supplementary file 1*). All 96 cell populations exposed to paraquat adapted much faster than every other cell population exposed to any other stressor (*Figure 1A*). Between 4 and 10 cell generations, populations on average reduced their doubling time by 106 min. Assuming the minimum achievable cell doubling time to be that of the wild type before exposure to paraquat (a mean of 93 min), this corresponded to 49.3% of the maximum possible reduction. Thenceforth, adaptation entered a second phase where the reduction in cell doubling time progressed much slower until it plateaued after 75 generations at 72.6% of the maximum possible reduction (mean) (*Figure 1A*).

We used a numerical model of the adaptation process to test if the extraordinarily fast paraquat adaptation could reasonably be accounted for by cell populations accumulating loss-of-function point mutations or chromosome duplications, both of which have previously been linked to fast Darwinian adaptation in experimental yeast populations exposed to selection (*Voordeckers and Verstrepen, 2015*). We combined population genetics and population dynamics theory with existing data on yeast mutation rates and our experimental measures of effect sizes of loss-of-function mutations and aneuploidies (*Gjuvsland et al., 2016*). The numerical model was completely unable to reproduce the observed extraordinarily swift response to paraquat, indicating that nuclear mutations were unlikely to explain the phenomenon (*Figure 1—figure supplement 2*).

We then tested experimentally whether a Darwinian adaptive process driven by selection of new mutations could account for the observed paraquat adaptation in a stress-release experiment. To this end, we exposed new cell populations to paraquat over many consecutive growth cycles. After each growth cycle, a fraction of the adapting cells was placed in a paraquat-free medium for 1–10 growth cycles before being exposed to paraquat once more. The rationale being that if the adaptation is due to accumulation of random mutations, loss of the adaptation would progress gradually and take many growth cycles. All 96 cell populations retained their acquired tolerance to paraquat (mean reduction in cell doubling time: 106 min) for only 1–3 growth cycles before abruptly losing it (*Figure 1B*). When employing the same experimental procedure to 96 cell populations from each of the two other environments to which adaptation was also fast (arsenic and glycine), we found that despite the presence of a much stronger Darwinian counterselection (*Figure 1C*), these populations lost their acquired adaptations more slowly and gradually (*Figure 1B*). Thus, while a Darwinian mutation/selection-based adaptive process could potentially explain the data for seven of the eight tested stressors, the paraquat adaptation could hardly be reconciled with such a process.

## Mitophagy is not responsible for the swift first adaptation phase

As the fast adaptation was unlikely to be a result of canonical mutation/selection dynamics, we went on to investigate whether a mitophagic response was responsible. Mitochondrial fragmentation is a well-documented prelude to canonical mitophagy (*Sprenger and Langer, 2019*). We therefore first assayed mitochondrial morphology before, during and after paraquat exposure by confocal and electron microscopy. In both cases, paraquat caused a rapid shift (<5 hr) from a tubular to a fragmented mitochondrial organization (*Figure 2A and B*, *Figure 2—figure supplement 1*). After removing the paraquat stress we observed a rapid reversal (<5 hr) back to a tubular organization (*Figure 2B*). These results are consistent with the notion that mitochondrial $O_2^{.-}$ generation influences the mitochondrial fission and fusion dynamics (*Frank et al., 2012*; *Hung et al., 2018*; *Sprenger and Langer, 2019*). Nevertheless, the cumulative mitochondrial volume remained near pre-stress levels with at the most a marginal reduction after 77–79 hr of paraquat exposure (*Figure 2A and B*). Most importantly, cell populations (n=16) lacking Atg32, a key component of canonical mitophagy (*Liu and Okamoto, 2021*), adapted to paraquat over ~80 generations as wild-type populations (*Figure 2C*). This led us to conclude that the initial swift adaptation to paraquat did not depend on canonical mitophagy.

## mtDNA segmental deletions cause the swift adaptation to paraquat

After excluding canonical mitophagy we considered mtDNA copy number variation as a possible explanation for the swift adaptation. We first used short read sequencing to measure the mean coverage of the mitochondrial genome in five of the 96 yeast cell populations adapting to paraquat (*Figure 3A*). In all five populations, we found the mean mtDNA copy number to decrease dramatically during the early phase of paraquat adaptation. This led us to use qPCR to track changes in the copy numbers of individual mtDNA genes in nine paraquat-adapting cell populations. We found that all nine cell populations lost copy numbers of some, but not all, mtDNA-encoded genes during the early adaptation phase (*Figure 3B*, *Figure 3—figure supplement 1*). As adjacent genes were lost concomitantly, and to the same extent, the observed loss in copy numbers implied that the early adaptation phase was associated with deletion of entire mtDNA segments. In addition, the qPCR data showed that the lost segments were unevenly distributed across the mitochondrial genome: all nine cell populations lost one or more segments within the mtDNA region spanning *COX1* to *VAR1*, while a few also lost the 21 S rRNA and *COX2* rapidly thereafter. The lost segments also contained almost all of the mitochondrial tRNAs. The retained mtDNA segments, which in all nine cases encompassed *COX3-RPM1* and

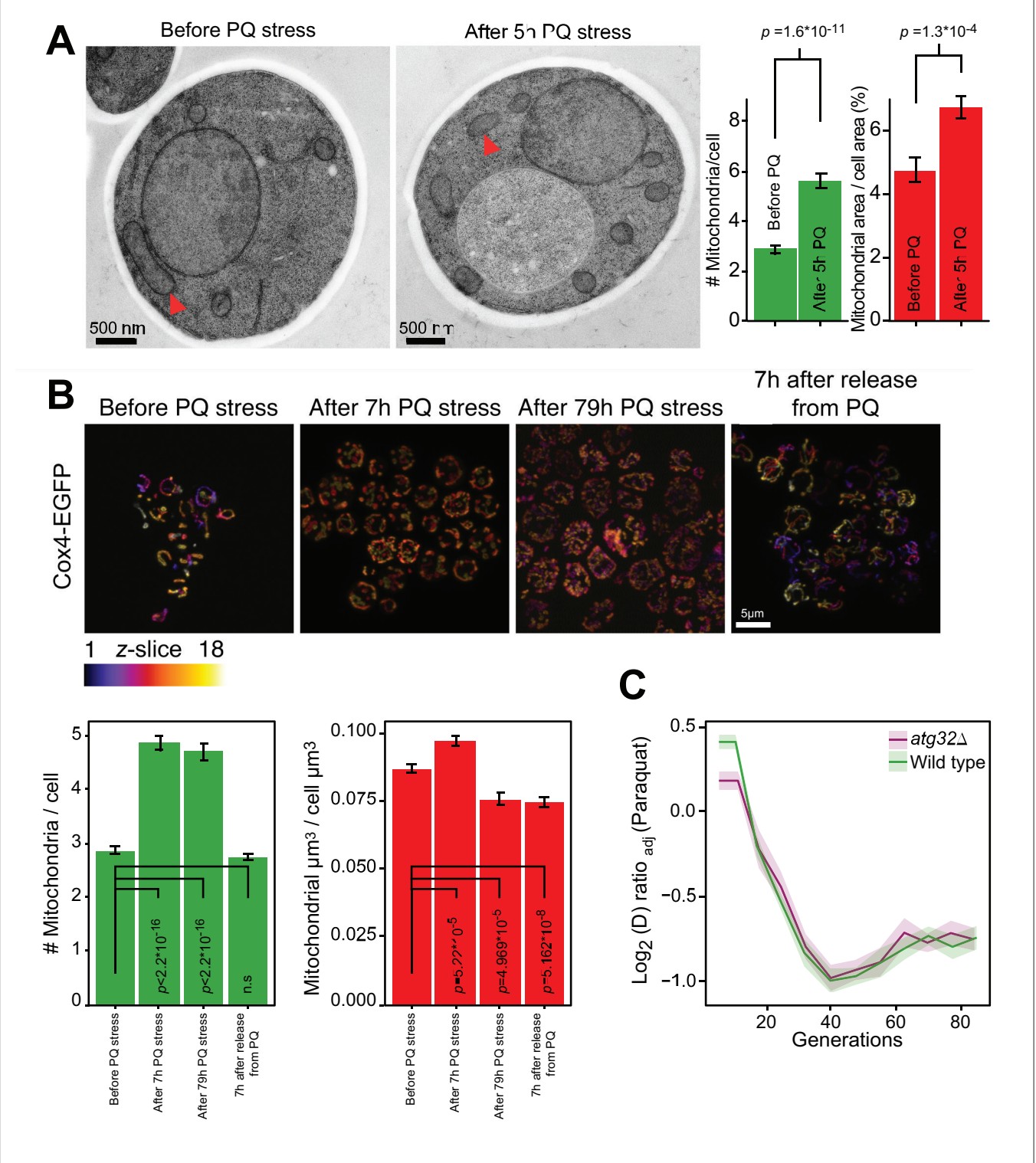

**Figure 2.** Mitochondrial fragmentation precedes the swift adaptation to paraquat (PQ). (**A**) EM microscopy of cells before (panel 1) and after short-term stress (panel 2, red arrowheads mark representative mitochondria). Panel 3 shows the number of mitochondria per imaged cell (left) and the imaged cell area occupied by mitochondrial area (%) (right), used as proxy for mitochondrial volume. Error bars: S.E.M. (n=100 cells). p-values: Welch two-sided t-test. (**B**) Confocal microscopy of cells with a Cox4-GFP mitochondrial tag. Color: z-dimension (yellow = front, purple = back; 18 slices). *Lower left diagram:* Number of mitochondria per imaged cell. *Lower right diagram:* Mean sum of mitochondrial volume as a fraction of cell volume. Error bars: S.E.M. (n=473–910 cells). p-values: Welch two-sided t-test (**C**) Adaptation of *atg32Δ* and wild-type populations to paraquat. Shade: S.E.M. (n=16 populations, each measured at n=6). See also *Figure 2—figure supplement 1*.

*Figure 2 continued on next page*

*Figure 2 continued*

The online version of this article includes the following source data and figure supplement(s) for figure 2:

**Source data 1.** Mitochondrial and cell area quantified, based on electron microscopy micrographs.

**Source data 2.** Number of mitochondria quantified, based on electron microscopy micrographs.

**Source data 3.** Number and volume of mitochondria of cells segmented from confocal microscopy micrographs of cells exposed to paraquat.

**Source data 4.** Doubling time data of wild type and populations adapting to paraquat over generations G; doubling times are in paraquat.

**Figure supplement 1.** Paraquat (PQ) stress leads to rapid mitochondrial fragmentation.

15 S rRNA, remained at near founder lever throughout the early adaptation phase. Since the mtDNA coverage prior to paraquat adaptation was perfectly even (*Figure 3—figure supplement 2A*), the observed mtDNA loss was clearly induced by paraquat.

As even small mtDNA deletions are prone to cripple oxidative phosphorylation (OXPHOS) (*Fontana and Gahlon, 2020*), we predicted that the observed loss of multiple protein, rRNA and tRNA functions had caused a substantial loss of OXPHOS. We therefore tested the respiratory growth capacity of the adapted populations by growing them on glycerol and found that the early paraquat adaptation coincided almost perfectly with loss of respiratory growth (*Figure 3C*). Similarly, the rapid loss of the acquired adaptations after removal of the $O_2^{\cdot-}$ stress coincided with the restoration of the capacity for respiratory growth (*Figure 3—figure supplement 2B*). To ensure that this restoration was associated with restoration of the mtDNA pool, we repeated the stress release experiment on the five sequenced populations and tracked, by short-read sequencing, the change in copy numbers of the lost mtDNA segments over the course of the experiment. As expected, the five populations quickly restored their capacity for respiratory growth after removal of paraquat, and this restoration coincided with the reestablishment of the wild type mtDNA profile and the loss of the acquired paraquat adaptation (*Figure 3D*, *Figure 3—figure supplement 2B, C*).

Together with the observation that $\rho^0$ yeast cells devoid of mtDNA devoid of OXPHOS growing on a normal glucose medium show a twofold decrease in cellular $O_2^{\cdot-}$ production (*Reddi and Culotta, 2013*), the data led us to hypothesize that the reduction of paraquat toxicity, leading to a substantial reduction in cell doubling time during the first adaptation phase, was closely associated with the loss of OXPHOS caused by loss of critical mtDNA segments.

We therefore predicted that cells already deprived of mtDNA would be preadapted to paraquat, i.e. their growth rate would be on par with that of cells that had just gone through the initial swift adaptation. To test this, we deleted the sole mitochondrial DNA polymerase Mip1 (Pol γ homolog) (*Lodi et al., 2015*), and cultivated the resulting $\rho^0$ cell populations (n=12) in the presence and absence of paraquat. In the absence of paraquat, the growth of *mip1Δ* cells was slower than that of the wild type cells (mean doubling time, *mip1Δ*=3.54 hr, WT = 1.55 hr). However, they did not show the pronounced petite phenotype (very small colonies) that characterizes *mip1Δ* lab strains, and which is caused by defects in the mitochondrial amino acid biosynthetic machinery (*Vowinckel et al., 2021*). Thus, the $\rho^0$ state was not likely to have spurred major compensatory cellular reconfigurations (*Veatch et al., 2009*) that could possibly have caused a response to paraquat distinctly different from that of the wild type. The *mip1Δ* cells were clearly preadapted to paraquat as their growth was equivalent to that of wild-type populations after these had undergone nine generations of adaptation to paraquat (*Figure 3E*). Together with the finding that the adaptation rate of *mip1Δ* populations exposed to paraquat was on par with the adaptation rate of wild-type cells in their second phase of adaptation (*Figure 3—figure supplement 3A*), this implies that loss of OXPHOS due to loss of mtDNA segments was indeed the most important factor underlying the first adaptation phase (*Figure 1A*).

To further test the validity of the *mip1Δ* results in different genetic backgrounds, we exposed the >4.700 gene knockout strains in the lab strain BY4741 yeast deletion collection to paraquat. Genes encoding mitochondrial proteins were overrepresented in the 100 most paraquat-resistant deletion strains (32.3% vs 18.6%, permutation test, p=0.00314). We also considered all gene deletion strains recently reported to have a low mtDNA copy number (mean < 1 copy per cell) (*Puddu et al., 2019*; *Grant et al., 1997*). As a group, these strains convert a cell doubling time defect in absence of paraquat (mean 16 min slower than deletion strains with a normal mtDNA copy number of 5–35 copies, p=3.05*10⁻⁶) into a cell doubling time advantage in presence of paraquat (mean 13 min faster than deletion strains with a normal mtDNA copy number of 5–35 copies, p=0.045). In

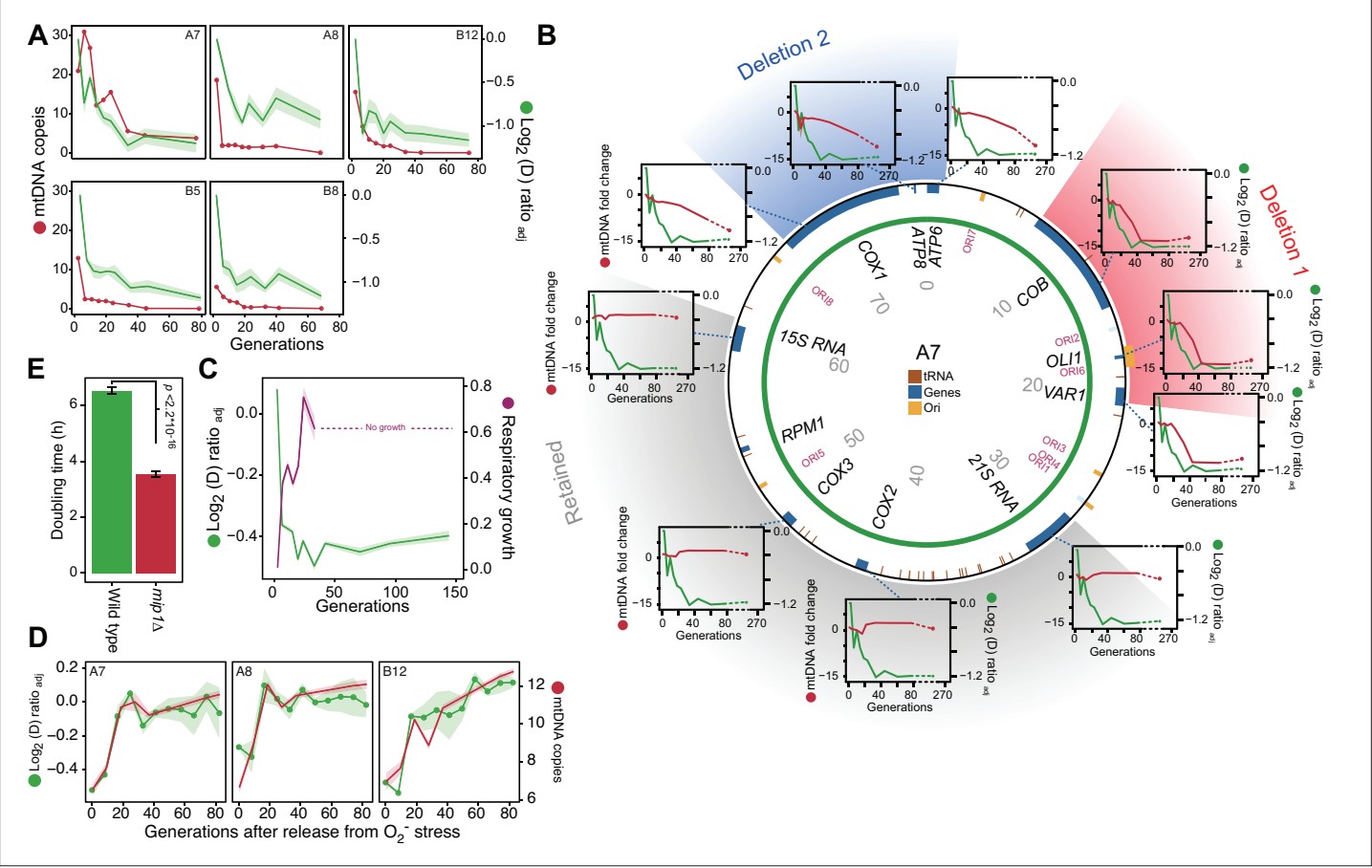

**Figure 3.** mtDNA editing causes the early adaptation to paraquat. (**A**) mtDNA copy number change (left *y*-axis, red line, median coverage relative to the haploid nuclear genome) during paraquat adaptation (right y-axis, green line, n=6) for 5 populations (panels). Shade: S.E.M. (**B**) mtDNA deletions associate with the paraquat adaptation. *Circle:* mtDNA (77 kb) before exposure to paraquat. Genes, origins of replication and position (kb) are indicated. *Coloured fields:* mtDNA deletions with concerted copy number change. *Diagrams:* mtDNA copy number change (left *y*-axis, purple line) of individual mtDNA genes during adaptation (right *y*-axis, green line) in population A7. Shade: S.E.M. (n=3). (**C**) The early-phase paraquat adaptation coincides with the loss of respiratory (glycerol) growth. Shade: S.E.M. 96 populations, each measured at n=5. Broken line indicates no growth (cell doubling time >24 h). (**D**) Recovery of the copy number of deleted mtDNA (right *y*-axis, red line) after release from 6 generations of paraquat exposure coincides with loss of the early-phase paraquat adaptation (left *y*-axis, green line, shade = S.E.M (n=15)) in populations A7, A8, and B12. (**E**) Doubling time (h) of wild type and *mip1Δ* cell populations in paraquat. Error bars: S.E.M. (n=191). p-values: Welch two-sided t-test. See also *Figure 3—figure supplements 1–3*.

The online version of this article includes the following source data and figure supplement(s) for figure 3:

**Source data 1.** Doubling time data of 96 populations adapted to paraquat for *G* generations exposed; doubling times are in paraquat and respiratory media (glycerol).

**Source data 2.** Mean log₂ coverage of 1 kb windows spanning the mitochondrial genome of five sequenced paraquat adapting populations over generations *G* of selection.

**Source data 3.** Mean log₂ coverage of 1 kb windows spanning the mitochondrial genome of sequenced populations adapting to paraquat and then released from this selection; data is given as a function of generations *G* of relaxation of selection.

**Source data 4.** qPCR data for mitochondrial DNA genes and nuclear DNA controls over generations of paraquat adaptation.

**Figure supplement 1.** Editing of mtDNA during the early adaptation to paraquat.

**Figure supplement 2.** Homeostatic restoration of mtDNA copy numbers and ability for respiration after release from paraquat (PQ) stress.

**Figure supplement 2—source data 1.** Doubling time data of populations adapted to paraquat for *Gₛ* generations and then released from selection for *Gᵣ* generations; doubling times are in paraquat and respiratory media (glycerol).

**Figure supplement 3.** Control of paraquat resistance through mtDNA deletions.

**Figure supplement 3—source data 1.** Doubling time data of 96 populations adapted over G generations to paraquat; doubling times are in 0 and 0.25 mM of menadione.

addition to being fully consistent with the *mip1Δ* data, these results demonstrate the considerable phenotypic penetrance of mtDNA loss across a range of genetic backgrounds and perturbed cellular physiologies.

Contemplating the possibility that mtDNA loss does not drive paraquat resistance by crippling OXPHOS, but causes enhanced cellular exclusion or inactivation of paraquat by some unknown mechanism specific to paraquat, we next exposed wild type populations, at different stages of paraquat adaptation, to menadione. As menadione is a mitochondrial $O_2^{\cdot-}$ generator that is structurally distinct from paraquat (*Fukui et al., 2012*), paraquat adapted cells should in this case not be preadapted to menadione. However, they were strongly preadapted to menadione, and this preadaptation became manifest during the early paraquat adaptation, that is concomitant with the mtDNA loss (*Figure 3—figure supplement 3B*).

## The deletion of mtDNA segments requires *SOD2*

Together, the above results strongly implied that mtDNA deletion was the predominant mechanism underlying the first adaptation phase. However, the data did not allow any firm judgement of whether the deletions were under regulatory control or were just due to unspecific paraquat-induced oxidative damage. We reasoned that if mtDNA deletion was part of a deliberate regulatory scheme for handling supraphysiological $O_2^{\cdot-}$ production, then deletion of genes coding for proteins being key for this regulation would delay the adaptation. On the other hand, if the mtDNA deletions were just due to paraquat-induced unspecific oxidative damage there would be no such delay. Because paraquat exposure induces supraphysiological production of $O_2^{\cdot-}$, we tested these mutually exclusive outcomes by deleting the two key actors in mitochondrial redox sensing and signaling, the $O_2^{\cdot-}$ dismutases Sod1 and Sod2 (*Zou et al., 2017*; *Reddi and Culotta, 2013*).

Cells lacking either one of the $O_2^{\cdot-}$ dismutases showed virtually no growth when exposed to the original paraquat dose (400 µg/mL) (*Figure 4—figure supplement 1A*), demonstrating the importance of enzymatic $O_2^{\cdot-}$ dismutation. The kinetics of adaptation depends heavily on the strength of selection (*Couce and Tenaillon, 2015*) in the sense that the adaptation rate is positively correlated with the stress level as long as the selection pressure is not overwhelming. In the case of paraquat, it was recently shown that 335 single gene deletion strains adapted near exactly as fast as predicted by the level of stress they experienced (*Persson et al., 2022*). In order for the stress level, that is the cell doubling time, of the mutant cells to be comparable with that of the wild type at 400 µg/mL, and thus provide a relevant deletion test, we reduced the paraquat concentration to 12.5 µg/mL. At this concentration, we found that the *sod1Δ* populations, in terms of reduction in cell doubling time, adapted as the wild-type populations over 10 growth cycles, while the *sod2Δ* populations barely showed any adaptive response (*Figure 4A*).

To exclude that the lack of any adaptive response in the *sod2Δ* populations at 12.5 ug/mL was because the actual stress level was lower than anticipated such that the mtDNA deletion response was not triggered, we exposed eight *sod2Δ* populations to a paraquat concentration of 50 µg/mL. Four populations still showed very marginal adaptation and we confirmed by qPCR that their mtDNA gene copy number remained at, or near, pre-stress levels (*Figure 4B*). The remaining four populations had a very delayed adaptation that coincided with different, single mtDNA deletions (*Figure 4B*, *Figure 4—figure supplement 1B*). To exclude that the delay in adaptive response was due to a lower selection for mtDNA deletions shutting down OXPHOS in the *sod2Δ* populations, we deleted Mip1 in the *sod2Δ* background. In 50 µg/mL of paraquat there was a marked reduction in doubling time in *sod2Δmip1Δ* populations relative to *sod2Δ* populations. The difference was similar to the one between *mip1Δ* populations and wild type populations in 400 µg/mL of paraquat (*Figure 4C*). This implies that the delayed adaptation in *sod2Δ* strains was because the mtDNA deletions causing shutdown of OXPHOS emerged at a much lower rate than in the wild type.

While Sod2 is located in the mitochondrial matrix, Sod1 is located in the cytosol and mitochondrial intermembrane space. The fact that *sod1Δ* populations adapted as the wild type suggests that the location of the redox signaling involved in the triggering of the mtDNA deletion response is the mitochondrial matrix. Regardless of which signals and processes within the matrix that are responsible for inducing and propagating down-stream effects, the above results strongly suggest that without Sod2 the cells appear incapable of launching the speedy mtDNA deletion process we observe in the wild type.

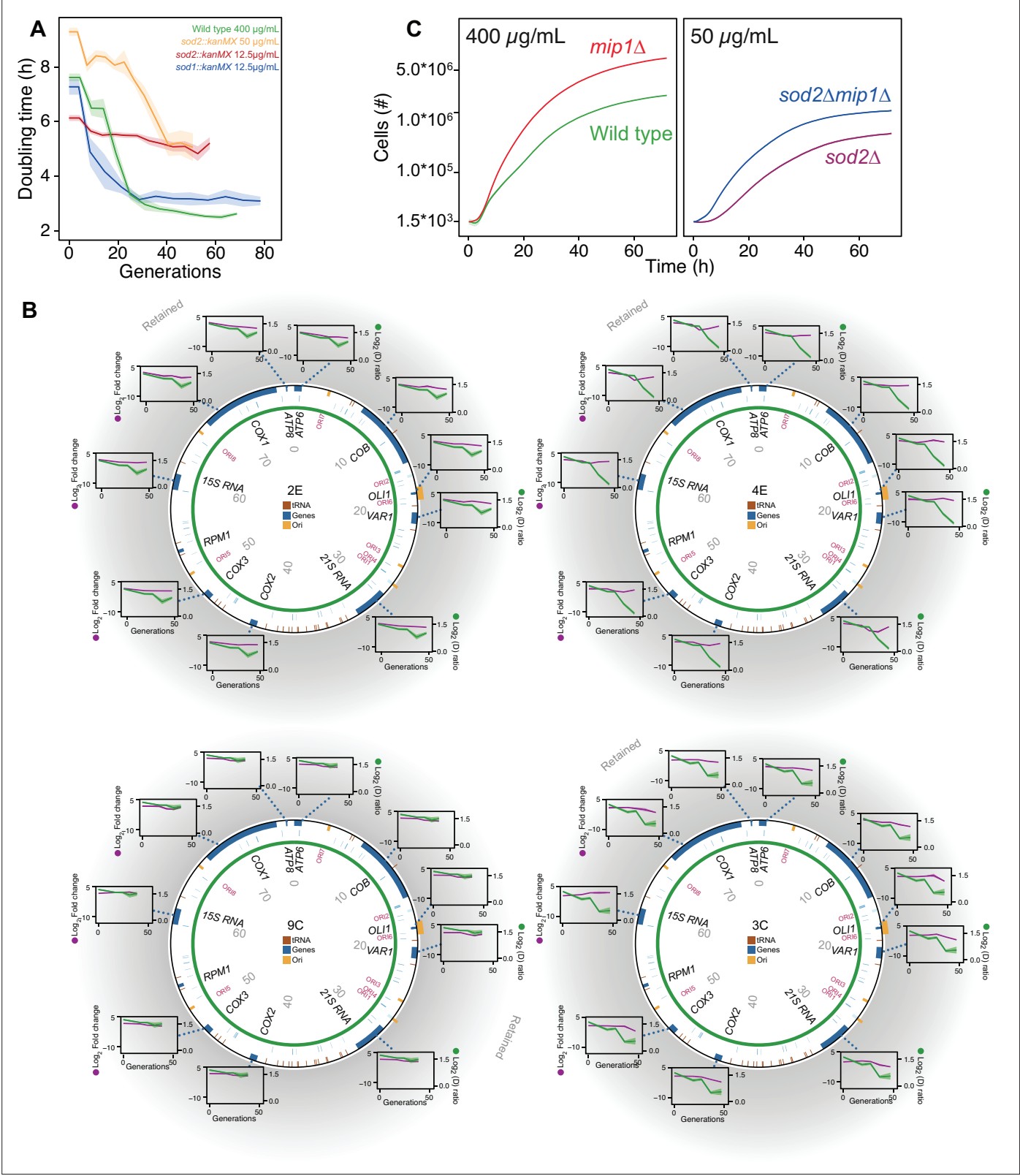

**Figure 4.** The mtDNA editing critically involves Sod2. (**A**) Doubling time (h) in wild type (400 µg/mL paraquat: green), *sod2Δ* (12.5 µg/mL paraquat: red; 50 µg/mL paraquat: yellow) and *sod1Δ* (12.5 µg/mL paraquat: blue) cell populations adapting to equivalent stress levels. (**B**) mtDNA change in *sod2Δ* cell populations 2E, 4E, 9C, and 3C adapting to 50 µg/mL paraquat. *Circle:* mtDNA (77 kb) before exposure to paraquat. Genes, origins of replication and nucleotide positions (kb) are indicated. *Colored fields:* mtDNA deletions with concerted copy number change. *Diagrams:* mtDNA copy number

*Figure 4 continued on next page*

*Figure 4 continued*

change left *y*-axis, purple line, (n=2) of individual mtDNA genes during the adaptation (right *y*-axis, green line). Shade: S.E.M. (**C**) Mean growth of wild type (n=480; green), *mip1Δ* (n=288; red), *sod2Δ* (n=96; purple), and *sod2Δmip1Δ* (n=384; blue) cell populations in the presence of 400 µg/mL (left) and 50 µg/mL (right) paraquat. See also **Figure 4—figure supplement 1**.

The online version of this article includes the following source data and figure supplement(s) for figure 4:

**Source data 1.** Doubling time data of wild type, *sod2Δ*, and *sod1Δ* populations adapting to paraquat; doubling times are in paraquat.

**Source data 2.** Growth curves of populations of *mip1Δsod2Δ*, *mip1Δ*, *sod2Δ*, and wild type exposed to paraquat.

**Source data 3.** qPCR data for mitochondrial DNA genes and nuclear DNA controls in *sod2Δ* populations over generations of paraquat adaptation.

**Figure supplement 1.** The mtDNA deletion process critically involves Sod2, but not Sod1.

**Figure supplement 1—source data 1.** Mean growth curves of wild type, *sod1Δ*, and *sod2Δ* and wild type exposed to different concentrations of paraquat.

## The mtDNA editing process requires anterograde mito-nuclear communication

In budding yeast, deletion of mtDNA alters the expression of a multitude of genes resulting in increased glycolytic production of ATP (**Epstein et al., 2001**), dubbed the retrograde response. Upon mitochondrial OXPHOS dysfunction, the cytosolic protein Rtg2 (**Sekito et al., 2000**) causes the transcription factors Rtg1 and Rtg3 to translocate from the cytosol to the nucleus where they together activate retrograde transcription (**Rothermel et al., 1995**; **Rothermel et al., 1997**). This mechanism is the most well-documented channel in yeast for communicating mitochondrial dysfunction to the nucleus, and in particular mtDNA deletion (**Guaragnella et al., 2018**). However, activation of retrograde transcription has also been reported to cause mtDNA deletions (**Farooq et al., 2013**). We therefore probed whether the retrograde response is required for paraquat-induced mtDNA deletions by exposing *rtg2Δ* and *rtg3Δ* cell populations (n=16) to paraquat for 80 generations. In both cases, all cell populations failed to adapt (**Figure 5A**), and their capacity for respiratory growth was virtually unperturbed at the end of the experiment (**Figure 5B**).

Importantly, the growth of the *rtg2Δ* and *rtg3Δ* populations was similar to that of the wild type in the presence as well as the absence of paraquat. This implies that the lack of these proteins did not cause reduced growth by affecting the cellular growth physiology, such as anaplerosis. Considering the paraquat concentration used, the lack of adaptation after 80 generations of selection and the retainment of OXPHOS, the $O_2^{\cdot-}$ production was likely on par with the production in the wild type immediately after exposure to paraquat. This puts a very restrictive upper bound on the frequency of mtDNA deletions in the wild type that are caused by unspecific oxidative damage due to paraquat-induced increase in $O_2^{\cdot-}$ production. Thus, the data strongly support the notions that the mtDNA deletion process is under regulatory control and that this regulation is dependent on a two-way mito-nuclear communication facilitated by Rtg2 and Rtg3.

## Sustained mtDNA deletion causes irrevocable mitochondrial impairment

We found that 44 of the 96 populations had completely lost their capacity for restoring the pool of intact mtDNA genomes back to pre-stress levels after 24 generations of paraquat exposure, and after 242 generations all did (**Figure 6—figure supplement 1**). Moreover, in the five sequenced cell populations, the capacity to restore the copy numbers of intact mtDNA and the respiratory growth after removal of stress was lost between 15 and 42 generations of paraquat exposure (**Figure 6A**). Together, these data suggested that the deletion of mtDNA genes continued after the first adaptation phase, ultimately leading to a complete loss of intact mtDNA genomes. We therefore assayed the mtDNA loss in 44 sequenced endpoint ($t_{50}$) populations. Out of these, 25 populations had almost completely lost their entire 77 kb mtDNA (97–99%), retaining only small (<1 kb) segments (**Figure 6B**, **Figure 6—figure supplement 2A, B**), in line with previous reports (**Fangman et al., 1989**). 18 endpoint populations remained in a rho negative ($\rho^-$) state, retaining 6–34 kb mtDNA segments with copy numbers somewhat above the original founder levels (mean: 20% increase). The mtDNA genes in the region spanning *COX1* to *VAR1* were lost in all, or almost all, 18 populations, while *COX3-RPM1* and 15 S rRNA were retained in most populations, confirming the deletion bias observed in the early

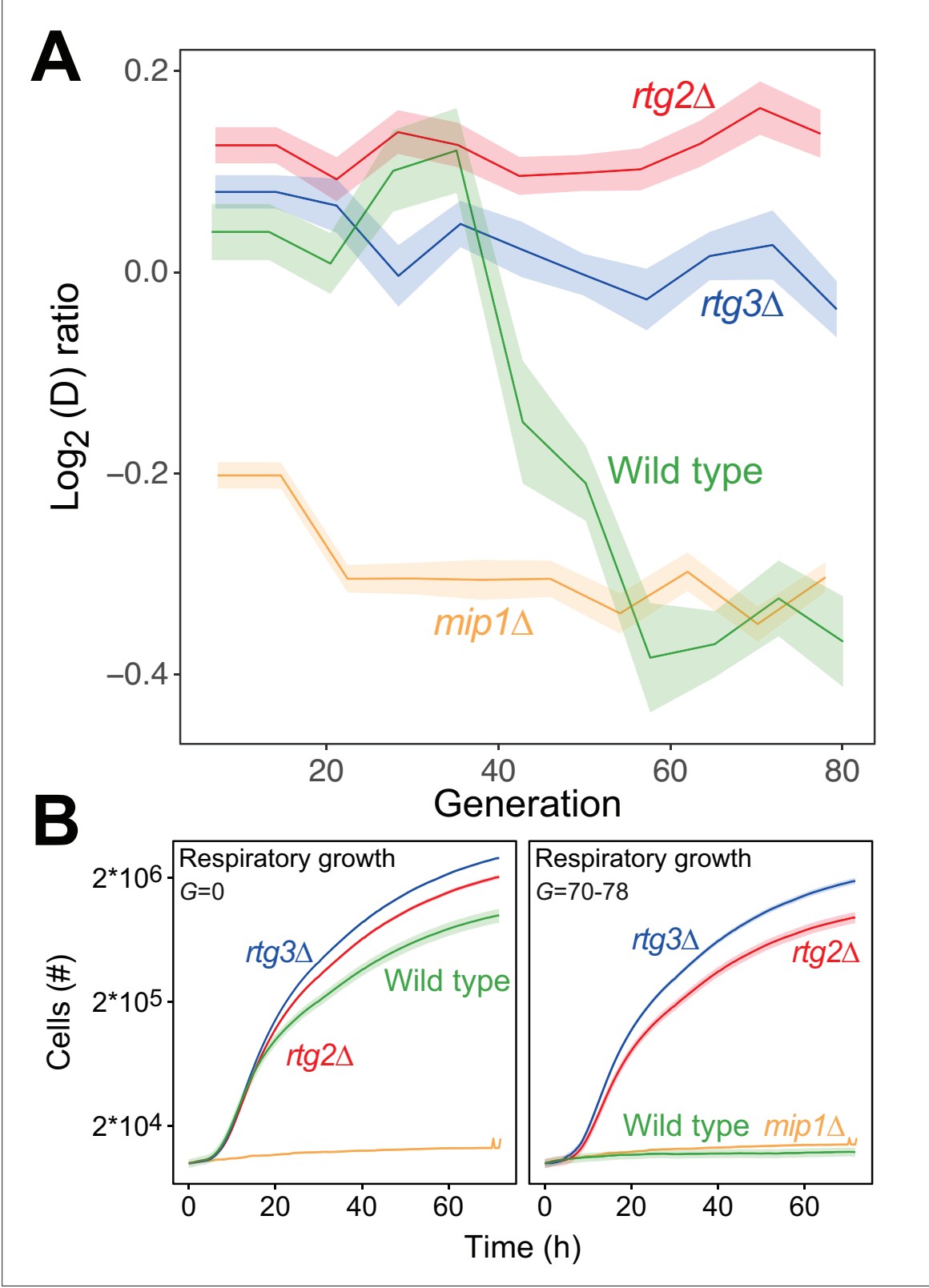

**Figure 5.** The mtDNA editing critically involves anterograde mito-nuclear communication. (**A**) Doubling time adaptation of 18 wild type, *rtg2Δ*, *rtg3Δ*, and *mip1Δ* cell populations to 400 μg/mL of paraquat. Shade: S.E.M. Each population type measured at n=4. (**B**) Respiratory (glycerol) growth of wild type, *rtg2Δ*, *rtg3Δ*, and *mip1Δ* cell populations, before (left) and after (right) 70–78 generations of paraquat adaptation. Shade: S.E.M (n=72–144 populations, each measured at n=1).

*Figure 5 continued on next page*

*Figure 5 continued*

The online version of this article includes the following source data for figure 5:

**Source data 1.** Doubling time data of wild type, *rtg2Δ*, *rtg3Δ*, and *mip1Δ* populations adapting to paraquat; doubling times are in paraquat.

**Source data 2.** Growth curves of wild type, *rtg2Δ*, *rtg3Δ*, and *mip1Δ*, adapted and not adapted to paraquat; doubling times are in respiratory media (glycerol).

adaptation phase (*Figure 6C*). The coverage across the retained mtDNA segments was even in each of these 18 $\rho^-$ endpoint populations, the most likely explanation being that it reflected a single continuous stretch of mtDNA (*Figure 6—figure supplement 2A*). To confirm this, we sequenced two end point populations (D1 and A7) with longer Nanopore reads. As expected, we found that many reads spanned the entire set of retained segments (*Figure 6—figure supplement 3*). This speaks strongly against a model where oxidative damage destabilizes the mtDNA and where parts of it are retained as different small fragments in different mitochondria and cells. The alignment data also suggested that some retained mtDNA segments persisted as linear tandem amplifications. To confirm this, we performed a PCR directed outwards across the ends of the retained mtDNA segments. This would produce a product only if two copies of the mtDNA segments were located next to each other on the same mtDNA molecule (*Figure 6—figure supplement 2C*). The PCR resulted in the expected product in the D1 endpoint population, showing that at least some of the mtDNA molecules in this population must therefore have been tandemly amplified, in line with the standard configuration of intact mtDNA in yeast cells (*Maleszka et al., 1991*).

To assess whether there was an adaptive advantage associated with the sustained depletion of mtDNA, we compared the doubling times of the $\rho^-$ populations still retaining 6–34 kb mtDNA segments with those that had lost almost all their mtDNA. Intriguingly, the latter group consistently grew slower on paraquat than the former (*Figure 6D*). This suggests that the sustained mtDNA depletion under long-term stress, leading to irreversible loss of intact mtDNA genomes, is an artifactual response driven by prolonged induction of a regulatory mechanism dimensioned by natural selection to handle $O_2^{\cdot-}$ stress it can successfully deal with before the pool of intact mtDNA genomes disappears.

## Chromosome duplications explain the second adaptation phase

Wild type cells realized 23.3% of their adaptation potential in the second adaptation phase (*Figure 1A*). This may, for example, be adaptation to the effects of mtDNA loss as such (*Figure 6—figure supplement 4A*), or to paraquat-induced $O_2^{\cdot-}$ production not dependent on the presence of an intact mtDNA pool (e.g. through association with the cytosolic Yno1, *Rinnerthaler et al., 2012*). In any case, the much slower doubling time reduction characterizing this second phase suggested that a Darwinian mutation/selection process was involved (*Figure 1—figure supplement 1*).

To search for nuclear genome changes that could explain the second adaptation phase we analyzed the sequence data of 44 random endpoint ($t_{50}$) populations. We did not find evidence for adaptive point mutations, that is mutations rarely occurred in the same gene across populations or coincided with growth improvement (*Figure 6—figure supplement 4B*, C). However, all but four endpoint populations carried extra copies of chromosomes II (n=29), III (n=21) and/or V (n=16) (*Figure 6—figure supplement 5A*) at near fixation (mean *p*: 0.97). In the five sequenced populations for which we had time-resolved data, these chromosome gains appeared after the early and very swift $O_2^{\cdot-}$ adaptation phase (*Figure 6—figure supplement 5B*). To assess their contribution to the second phase of adaptation, we crossed clones carrying the individual aneuploidies back to wild type cells over three consecutive meiotic generations. We then compared the tolerance for paraquat, and the capacity for respiratory growth, in offspring with and without extra chromosomes. Part of the paraquat resistance and part of the loss of respiratory growth co-segregated with chromosomes II and V aneuploidies. The duplications of chromosome II and V caused a reduction in respiratory growth relative to the wild type (cell doubling times: 5.7 hr (chr II), 4.9 hr (chr V), 3.3 hr (WT)) (*Figure 6—figure supplement 5C*). They also reduced the cell doubling time during paraquat exposure by 31 and 38 min, respectively (*Figure 6—figure supplement 5D*). Duplication of chromosome III caused no apparent reduction in cell doubling time. Assuming an additive phenotypic effect of the chromosome II and V duplications, the cell doubling time would be reduced by 69 min. Together with the effect from the mtDNA deletions disrupting OXPHOS function, this would correspond to cells realizing 81% of the possible

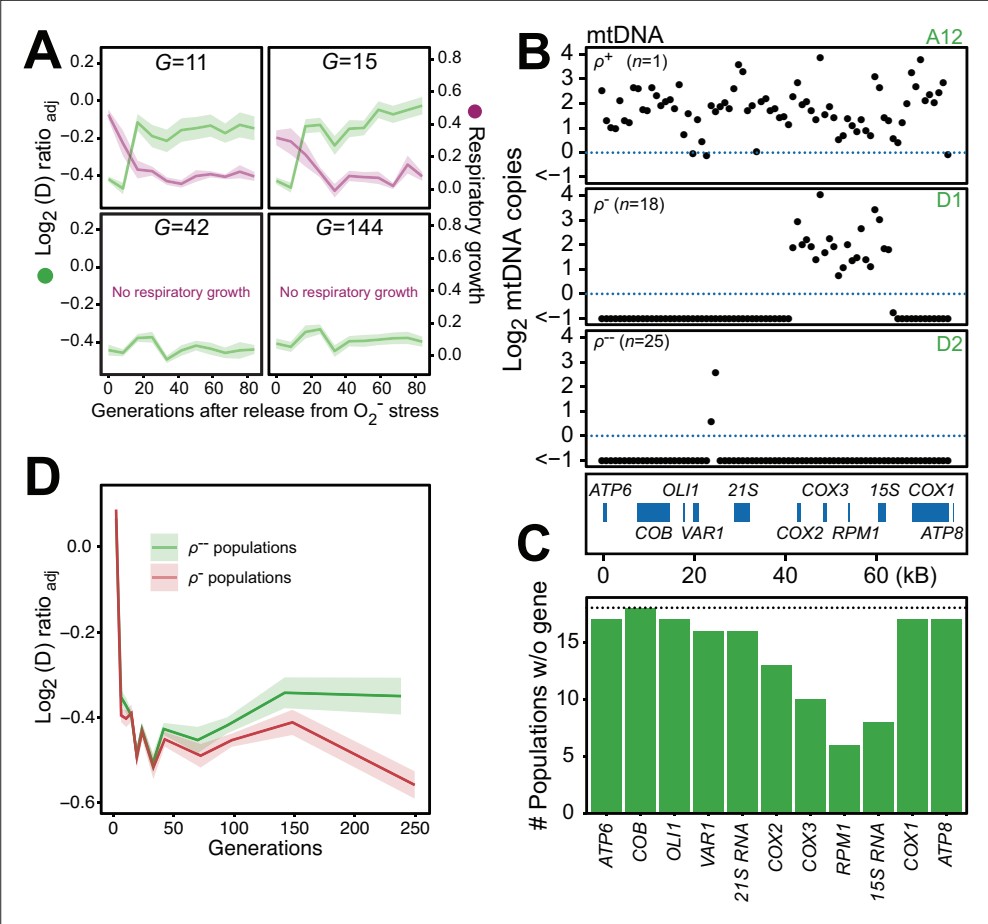

**Figure 6.** Chronic exposure to paraquat causes irreversible mitochondrial impairment by sustained mtDNA editing. (**A**) Paraquat adapting cell populations (*G*=generations of exposure to paraquat) ultimately lose their capacity to recover respiratory (glycerol) growth (right *y*-axis, purple line, $\log_2$ doubling time relative to founder) and the loss coincides with the genetic fixation of the paraquat adaptation (left *y*-axis, green line). Shade: S.E.M. of 5 populations, each measured at n=5. (**B**) All but one ($\rho^+$) sequenced cell population adapted to long-term paraquat stress ($t_{50}$) retain only small (6–30 kb; $\rho^-$) or very small (<2 kb, $\rho^{--}$) mtDNA segments. *Panels:* Representative populations. *y*-axis: mtDNA copy number (median coverage in 1 kb windows relative to haploid nuclear genome). Gene positions are indicated. (**C**) Number of $\rho^-$ populations after long-term paraquat exposure ($t_{50}$) in which the specified mtDNA gene was lost. (**D**) The $\rho^{--}$ populations became less fit than the $\rho^-$ populations during a long-term exposure to paraquat. See also *Figure 6—figure supplements 1–5*.

The online version of this article includes the following source data and figure supplement(s) for figure 6:

**Source data 1.** Mean $\log_2$ coverage of 1 kb windows spanning the mitochondrial genome of each sequenced paraquat adapted endpoint population.

**Figure supplement 1.** Long-term exposure to paraquat causes genetic fixation of adaptation.

**Figure supplement 1—source data 1.** Doubling time data of 96 populations adapted to paraquat for $G_s$ generations, followed by release from this selection over $G_r$ generations; doubling times on paraquat.

**Figure supplement 2.** mtDNA loss during long-term exposure to paraquat.

**Figure supplement 3.** Dot-plot mapping of long mtDNA reads from Nanopore sequencing of clones from the D1 and A7 paraquat adapted populations.

**Figure supplement 4.** Nuclear genome evolution during long-term exposure to paraquat (PQ).

**Figure supplement 4—source data 1.** Small indels and SNPs called in sequenced paraquat adapted endpoint populations.

**Figure supplement 4—source data 2.** Doubling time data of *mip1Δ* cells grown in stress, and of wild type cells grown in no stress.

*Figure 6 continued on next page*

*Figure 6 continued*

**Figure supplement 5.** Chromosome duplications explain the second phase of adaptation to paraquat.

**Figure supplement 5—source data 1.** Mean $\log_2$ coverage for each chromosome in each sequenced paraquat adapted endpoint 1 population.

**Figure supplement 5—source data 2.** Mean log2 coverage for each chromosome in five sequenced paraquat adapting populations over generations *G* of selection.

---

adaptation. As the cells actually realized on average 72.6%, an approximately additive phenotypic effect of these duplications appears to fully explain the second phase of adaptation in populations having both duplications genetically fixed.

## Discussion

As argued above, the lack of adaptation of the *sod2Δ*, *rtg2Δ*, and *rtg3Δ* populations are hard to reconcile with the operation of random $O_2^{-}$ induced mtDNA deletions (e.g. caused by genomic instability) and subsequent selection of cells possessing OXPHOS-impaired mitochondria. Instead, the data support the notion that the observed deletion of mtDNA segments causing loss of OXPHOS activity is under regulatory control. We think it is apt to denote this regulatory process as 'mtDNA editing' as it alters the mtDNA content to suit a particular purpose (*Merriam-Webster's Coll Thes, 2021*). However, the fidelity of the mtDNA deletion mechanism in terms of which mtDNA segments are lost in the first adaptation phase does seem to be moderate. Still, there was a clear preference for deletions within the *COX1-VAR1* region, and the segment containing *COX3, RPM1,* and 15 S RNA were not deleted in this phase, and more rarely also in the second adaptation phase.

Any mtDNA deletion that removes an enzymatic function required for electron transfer at, or before, the point of electron leakage to oxygen, such as the *COB*-encoded cytochrome B in the cytochrome c reductase complex, or any of the rRNA or tRNA genes that are essential to expression of these functions, would be sufficient to interrupt the electron transfer to oxygen. Thus, the moderate specificity of the mtDNA deletion mechanism makes sense from an evolutionary point of view: natural selection would not be able to increase the fidelity of the mtDNA editing program beyond the point where no further adaptation is achieved. But it should be noted that all our paraquat adapting cell populations lost either *COB* itself, or genes required for *COB* expression, during the first adaptation phase. It cannot be excluded that deletions of some individual genes, e.g. of *COX1*, may cause increased $O_2^{-}$ production through mechanisms driven by paraquat or by excessive natural leakage of electrons to oxygen. But such single gene deletions cannot be present in our material for the simple reason that because of their negative effect on growth rate they would be exposed to purifying selection. That is, the gain in growth rate during the first adaptation phase is so huge that cells with a reduced growth rate due to increased $O_2^{-}$ production would not be able to propagate.

The speedy restoration of the wild type mtDNA pool after release from paraquat is most probably due to proliferation of heteroplasmic cells that become homoplasmic through recovery of their wild type mtDNA profile. However, the lack of selective advantage for the wild type mtDNA (*Figure 1B and C*) implies that this recovery is not caused by a Darwinian process based on differences in cell doubling time. Thus, either the replication of the mtDNA genomes possessing deletions has to stop (*Jakubke et al., 2021*; *Zhang et al., 2019*; *Chen et al., 2020*), or the standing pool of these mtDNA genomes specifically has to be removed by some sort of targeted mitophagy (*Twig et al., 2008*; *Ban et al., 2017*) . Both processes may also be simultaneously operative. Regardless of the finer details of how this restoration is orchestrated, one would expect that it is under homeostatic control.

Our data bring fresh perspectives to the table concerning (i) the relationship between stress-induced mitochondrial fragmentation and canonical mitophagy (*Zorov et al., 2019*), (ii) under which conditions do mitochondria deprived of OXPHOS genes produce more $O_2^{-}$ due to increased electron leakage (*Aerts et al., 2017*), (iii) whether clonal expansion is the main mechanism underlying the propagation of mitochondria containing deletions (*Nido et al., 2018*), and (iv) under which conditions, and to which degree, does the main retrograde response to mtDNA deletion in yeast mediate two-way mito-nuclear communication (*Guaragnella et al., 2018*). They also support the emerging notion that selective mitophagy is an important mechanism for local mitochondrial repair (*Gustafsson and*

*Dorn, 2019*), and that selective mitophagy may be deliberately repressed while the cell experiences $O_2^{\cdot-}$ stress.

Our results strongly suggest that there is an additional genetically controlled defense layer against ROS induced damage in budding yeast, situated between the primary antioxidant defenses and mitophagy. Budding yeast and humans share a range of evolutionary conserved mechanisms concerning respiratory chain biology and mitochondrial quality control, including antioxidant enzymes and key proteins regulating mitophagy (*Montava-Garriga and Ganley, 2020*; *Kumar and Reichert, 2021*; *Barrientos, 2003*). As our experimental yeast strain has a fully intact OXPHOS system very similar to higher eukaryotes, it is reasonable to expect that important features of the disclosed mtDNA editing mechanism may also be evolutionary conserved. In light of this, the available data on the cancer therapeutics doxorubicin and cisplatin suggest that our main results may extend to post-mitotic cells. They both accumulate in mitochondria (*Kalyanaraman et al., 2002*; *Genc et al., 2014*), where they induce $O_2^{\cdot-}$ production through redox cycling (*Malhi et al., 2012*; *Song et al., 2017*). Their long-term administration cause oxidative injury to a variety of cells and tissues (*Song et al., 2017*; *Songbo et al., 2019*; *Moruno-Manchon et al., 2018*; *Ren et al., 2019*), and they both increase the frequency of mtDNA deletions (*Genc et al., 2014*; *Adachi et al., 1993*). Arrestment of OXPHOS activity by mtDNA editing, followed by homeostatic restoration of the pool of intact mitochondrial genomes, may therefore possibly be a generic eukaryotic adaptation to obviate a costlier mitophagic response in a variety of situations. If so, the biomedical implications in terms of the etiology of age-related disease and therapeutic opportunities appear to be noteworthy.

## Materials and methods
### Yeast cells
#### Founder strain
We used a single, haploid clone of the *S. cerevisiae* strain YPS128 (*MATα ura3::NatMX*-barcode *ho::HYGMX*) as the background genotype. YPS128 is a wild, oak isolate with a North American genome composition (*Liti et al., 2009*) whose respiratory capacity has not been impaired by domestication (*De Chiara et al., 2020*). In contrast to common lab-strains, it carries neither *HAP1* defects, impairing mitochondrial regulation, nor *MIP1* defects that lead to spontaneous mtDNA loss (*Gaisne et al., 1999*).

#### Deletion strains
We generated a $\rho^0$ YPS128 strain, lacking all mtDNA, by deleting *MIP1*. We also constructed gene deletion strains lacking *SOD2*, *SOD1*, *CCP1*, *ATG32*, *YAP1*, *RTG2* and *RTG3*. Genotypes: YPS128, *MATα*, *ura3::natMX*-barcode, *ho::hygMXΔ*, *genex::kanMX*, with each target deleted from start to stop codon. We also constructed a *mip1Δsod2Δ* double deletion mutant as *sod2::kanMX, mip1::URA3*. For all strains, two to three independent clones, verified by PCR to carry the deletion cassette and to lack the deleted gene at the target locus, were isolated and retained. The independent clones were used as replicates in all experiments. For yeast deletion collection experiments, we used the haploid BY4741 single gene deletion collection (*MAT**a**;his3Δ1;leu2Δ0;met15Δ0;ura3Δ0; genex::kanMX*) (*Giaever et al., 2002*), which was cultivated in absence and presence of each stressor. Collection size: n=4580, each cultivated at n=6. We report the doubling time data for this collection in Data S7. Primers are reported in *Source data 1*.

#### Aneuploidic strains
Cells with and without one extra chromosome II, III or V were generated by repeatedly (3 x) backcrossing clones from endpoint ($t_{50}$) populations carrying chromosome duplications to founder clones of the opposite mating type (*MAT**a** ho::HYGMX*). Each backcross was done on YPD (Yeast Peptone Dextrose) medium using haploids verified by qPCR to retain the chromosome duplication. Diploid hybrids were selected after three days of growth on solid minimum media (0.675% Yeast Nitrogen Base (CYN2210, ForMedium), 2% (w/v) D-Glucose, pH = 6–6.5 (NaOH), 2.5% agar) medium. They were let to sporulate overnight on solid 1% potassium acetate sporulation medium to generate recombined haploids. These were genotyped at the *URA* and *ho* locus and *ura- MATα* haploids were passed on to the next round of backcrossing. After three rounds of backcrossing, we selected *ura- MATα* haploids

with (n=2 clones) and without (n=2 clones) the chromosome duplication of interest and estimated their respective growth rates (n=6) in a completely randomized design on the media of interest. We compared the cell doubling time of clones with and without the respective chromosome duplication. Primers are reported in *Source data 1*.

## Cox4-EGFP fusion strain

To construct Cox4-EGFP fusion as a reporter for mitochondrial morphology, founder cells were transformed with PCR fragments of EGFP amplified from pYM27, with flanking regions homologous to *COX4.* Downstream of *COX4* we inserted kanMX as selection marker during transformation. Cox4 localizes to the mitochondrial inner membrane (*Zhu et al., 2019*). Primers are reported in *Source data 1*.

## Cell cultivation media

Except where otherwise stated, yeast strains were cultivated on a Complete Supplement Mixture medium (CSM medium; hereafter: "Background medium") composed of 0.14% Yeast Nitrogen Base (CYN2210, ForMedium), 0.50% $NH_4SO_4$, 0.077% Complete Supplement Mixture (CSM, DCS0019, ForMedium), 2.0% (w/w) glucose, pH set to 5.80 with 1.0% (w/v) succinic acid and 0.6% (w/v) NaOH. For solid medium cultivations, 2.0% (w/v) agar was added. For pre-cultures to glycine, isoleucine, citrulline and tryptophan selection environments, we modified the background medium to avoid confounding growth on the stored nitrogen (*Gutiérrez et al., 2016*) by replacing CSM by 20 mg/L uracil and by reducing the $NH_4SO_4$ concentration (30 mg N/L). Simple modifications of the background medium were made to generate four of the eight stressor environments:+0.8 µg/mL rapamycin,+400 µg/mL paraquat (methylviologen; N,N-dimethyl-4–4'-bipiridinium dichloride),+3 mM arsenic ([As III]; $NaAs_2O_3$),+62.5 mg/L citric acid. To generate the four other stressor environments, we replaced $NH_4SO_4$ in the background medium with 30 mg N/L of L-glycine, L-isoleucine, L-citrulline or L-tryptophan, together with 20 mg/L uracil. For the respiratory growth experiments, we replaced 2% glucose with 2% glycerol. For the menadione growth experiments, we added 0.25 mM menadione to the background medium. For the vitamin C experiment, we added 180 mM of ascorbic acid (vitamin C) to background medium with and without paraquat. We cast all solid plates 10–15 hr prior to use in PlusPlates (Singer Instruments, UK), on a level surface, by pouring 50 mL of selection medium in the same upper right corner of each plate. We removed excess liquid by drying plates in a laminar airflow in a sterile environment. We stored cells at –80 °C in 20% glycerol and cultivated them at 30 °C. Populations were subsampled and transferred to and from plates using robotics (ROTOR HDA, Singer Instruments Ltd, UK), at the indicated transfer format.

## Experimental evolution of cells

We single streaked and then expanded a single haploid YPS128 clone to moderate colony size (~2 million cells), sampled the colony (~50,000 cells) and expanded the sample until stationary phase (~2 million cells; 36 hr) in 5 mL of background medium. A subsample of these founder cells were stored. We poured a sample of the stationary phase culture on top of a solid plate (background medium) and allowed the lawn of cells to grow, again until stationary phase (72 hr). We then repeatedly sampled the lawn using 384 short pin pads to generate eight solid plates with 1,152 colonies each. These colonies served as pre-cultures ($t_{-1}$) to the first selection cycle of each of the eight selection environments. We expanded these pre-cultures on background, or nitrogen background, medium until stationary phase (~2 million cells; 72 hr). We transferred samples of the pre-culture with 384 short pin pads to experimental plates to generate the 1,152 populations to be evolved in each selection environments (*Supplementary file 1*). We then cycled all 8 × 1152 populations through 50 rounds of expansion until stationary phase (72 hr), subsampling and transfer to fresh plates, to produce $t_1$ to $t_{50}$. We evolved many (n=24–192, see figure legends) *mip1Δ, rtg2Δ, rtg3Δ, sod1Δ, sod2Δ,* and *atg32Δ* cell populations in a similar design, over a varying number of growth cycles. We consistently interleaved several wild type cell populations to serve as controls on the same plates.

## Establishing and cultivating frozen chronological records of cell populations

In parallel to the sampling of colonies for transfer to fresh plates, we systematically sampled a large subset of populations in each environment to generate a frozen chronological record of their evolution. For each of the eight selection environments, we systematically sampled (1,536 short pin pads) the same 96 populations at the end of growth cycles 0, 1, 2, 3, 4, 5, 7, 9, 12, 15, 20, 25, 30, 35, 40, 45, and 50, to generate a dense chronological adaptation record of 768 populations. We transferred the samples to a liquid selection medium (100 µL), expanded the populations until stationary phase (72 hr), added 100 µL of glycerol (final concentration: 20% (w/w)) and stored them at –80 °C. We thawed and re-suspended these frozen stocks, and transferred cells (96 short pin pads) to a solid background, or a nitrogen background medium. To generate a randomized design, we used the *randint* function in the Python package NumPy (version 1.15.4). We pre-cultivated cells until stationary phase (72 hr), sampled and transferred pre-cultures (1,536 short pin pads) to selection environments plates, interleaving (384 short pin pads) 384 separately pre-cultivated, wild type, founder controls among the evolving populations on each plate. We cultivated all 1536 cells populations until stationary phase (72 hr), while tracking their growth and adaption as described below. Using the same design, we also established and cultivated a frozen chronological record of paraquat-exposed *mip1Δ, rtg2Δ, rtg3Δ, sod1Δ, sod2Δ,* and *atg32Δ* cell populations.

We performed three distinct release-from-selection experiments, using the frozen chronological records as start point. First, we thawed, re-suspended, sampled and transferred $t_0$, $t_1$, $t_2$, $t_3$, $t_4$, $t_5$, $t_7$, and $t_{50}$ samples of the 96 frozen paraquat-adapting populations to no stress solid medium plates. We evolved these populations over ten growth cycles (~84 generations) on no stress plates, sampled each population at the end of each growth cycle and stored samples at –80 °C (as above) to create a chronological record of samples first adapted to paraquat for different time-periods, and then released from the paraquat selection, again for different time periods. We thawed, re-suspended, sampled, randomized and pre-cultivated (no stress) this second chronological record, and sampled and transferred stationary phase cells to paraquat selection plates. Second, to compare the kinetics of loss of paraquat adaptive gains to that of populations adapting to other challenges characterized by fast adaptation, we repeated (3x) the above selection relaxation experiment, including also those adapting to arsenic and glycine. We selected the time point in the chronological record where the populations had achieved 70–90% of their endpoint adaptation. We then thawed, re-suspended and sampled these stocks, expanded revived cells under relaxed selection for 10 growth cycles and created a frozen chronological record, which was revived, randomized, pre-cultivated and cultivated in the original stress, as above (n=5). For the glycine-adapting populations a nitrogen-limited background medium was used. Third, to compare the kinetics of loss of paraquat adaptation to that of the restoration of wild type mtDNA and of respiratory growth, we again repeated the release-from-paraquat experiment, but only for the five sequenced paraquat-adapting populations (A7, A8, B12, B5, and B8). Procedures were as above, but we replicated the experiment for each sample 3x and assayed both paraquat and respiratory (2% glycerol) growth at n=5 (randomization) for each replicate.

## Tracking cell growth and adaptation

### Counting cells in growing populations

We assayed the growth of cell populations in all experiments using the Scan-o-matic system (***Zackrisson et al., 2016***), version 1.5.7 (https://github.com/Scan-o-Matic/scanomatic.git; ***Zackrisson, 2019***). Cultivation plates were maintained undisturbed and without lids for the duration of the experiment (72 h) in high-quality desktop scanners (Epson Perfection V800 PHOTO scanners, Epson Corporation, UK) standing inside dark, temperature (30.0 C) and moisture controlled thermostatic cabinets with air circulation. We imaged plates at 20 min intervals using transmissive scanning at 600 dpi, identified the position of colonies and extracted intensities for pixels included in, and outside, each colony. For each colony, we estimated its sum pixel intensity as well as the median pixel intensity of the local background, subtracted the latter from the former and converted the remaining cell-associated pixel intensity to cell counts by using a pre-established calibration function, which had been obtained by estimating cell numbers using both spectrometry and flow cytometry. We smoothed and quality controlled growth curves, rejecting approximately 0.3% of growth curves as erroneous while being blinded to sample identities (for details, see ***Zackrisson et al., 2016***). To allow direct

visual comparison of growth curves of different samples while accounting for confounding effects from initial population size differences, we adjusted growth curves shown in figures in the $y$-dimension. We applied the function $N_{t,adjusted} = 2^{log_2(N_t)/[log_2(N_t)/log_2(median(N_0))]}$ to the mean growth curves to be visualized in figures, where $N_0$ is the mean initial population size across replicates, $N_t$ is the mean population size at time $t$ across replicates, and the median($N_0$) is the median of the mean $N_0$ of the samples to be visualized together.

## Cell doubling time and adaptation

We extracted the cell doubling time, $D$, from expanding cell populations. We used the 384 fixed spatial controls introduced at every fourth position to account for systematic doubling time variations within and across plates. By interpolating across the $log_2(D)$ values of the 384 measured controls (see *Zackrisson et al., 2016*) we estimated the $log_2(D)$ value a control colony would have had in each position. From the $log_2(D)$ value for each colony we then subtracted the corresponding $log_2(D)$ control value, thereby obtaining a normalized, relative $log_2$ doubling time, $log_2(D)_{norm}$. When relevant, we also adjusted the $log_2(D)_{norm}$ value for the bias associated with spatial controls having a slightly different pre-cultivation history than evolving populations, by use of the equation $log_2(D)_{adj} = log_2(D_t)_{norm} - log_2(D_0)_{norm}$, where the subscripts $t$ and $0$ refer to the growth cycle number. In some cases, we converted $D_{norm}$ back to a doubling time in hours while maintaining the normalization in order to ease interpretation. This measure is denoted *Doubling time* in figures, and set equal to $2^{Dnorm}D_{control, grand}$, where $D_{control, grand}$ is the grand mean of the raw doubling times of all controls run in a particular experimental series.

## Counting cell generations

We estimated the number of cell generations for any missing growth cycle by interpolating the values estimated for the two adjacent growth cycles. For each cell population, the total number of cell generations was calculated by summing over all growth cycles.

## Maximum possible reduction in cell doubling time

We estimated how much of the maximum possible reduction in cell doubling time the paraquat adapting populations had achieved at a given generation number by comparing their cell doubling times with that of the founder population growing on normal medium, assuming that the latter represented a lower boundary for what was physiologically possible.

## RNA sequencing to measure *SOD1*, *SOD2* and *CCP1* expression

Wild type cell populations were pre-cultivated for two consecutive 72 hr growth cycles on no stress background medium, sampled and transferred to background medium w. and w/o 400 µg/mL paraquat (as above). We exposed cells to paraquat for three growth cycles and then removed the paraquat for one additional growth cycle. We sampled cell populations: (i) immediately (10–15 s) after transfer (paraquat cycle 1 and 2), after 0.75 hr (paraquat cycle 1 and 2), 1.5 hr (paraquat cycle 1), 5 hr (paraquat cycle 1 and 2), 20 hr (paraquat cycle 1), and 25 hr (paraquat cycle 1). Cells to be harvested at early time-points (<5 hr after transfer) were cultivated in a 6,144 colony format, otherwise we used a 1,536 format. All samples corresponding to the same growth cycle were cultivated in parallel. To generate one replicate of one sample, we harvested all colonies on a plate by pouring 5 mL of liquid medium, w. or w/o paraquat, on top of the solid medium and scrapping off colonies with a sterile plastic rake into this liquid medium. The cells were pelleted at 12,000 G (2 min in 4 °C), re-suspended in RNAlater (Sigma Aldrich R0901) and stored at 4 °C. We extracted RNA from all the stored samples in parallel, first diluting the RNAlater solution with an equal volume of PBS and then pelleting cells at 5000 G (5 min, 4 °C). Cells were lysed by adding 600 µL of acid washed 0.5 mm beads and subsequent homogenization in a FastPrep homogenizer (three rounds at 40 s at 6 m s$^{-1}$ separated by 1 min on ice). RNA quality was determined using a Tapestation 2200 and Nanodrop (threshold; ABS$_{260/280}$ > 2.2 and RINe > 8). RNA sequencing was performed at SciLife (Stockholm, Sweden) using the Illumina TruSeq Stranded mRNA kit and a NovaSeq 6000 S4. RNA reads were checked for contamination using FastQ Screen (*Wingett and Andrews, 2018*). Filtered reads were aligned to the YPS128 reference genome using STAR (*Dobin et al., 2013*), and optical duplicates were marked with Picard-tools. The abundance of the *SOD1*, *SOD2* and *CCP1* transcripts was quantified with featureCounts from the subread

package across all samples (*Liao et al., 2014*). We normalized their read counts as fragments per kilobases per million reads, using the DESeq2 package for R (*Love et al., 2014*). We estimated significant differences compared to no stress at $t_0$ using Wald tests and Benjamini-Hochberg FDR correction, with a cut-off of $q<0.05$. The normalized read counts for *SOD1*, *SOD2,* and *CCP1* are reported.

## DNA sequencing of evolving cell populations

### Long read (PacBio) sequencing of the YPS128 founder strain

The total genomic DNA was extracted from a founder population cultivated overnight in background medium, using a standard phenol-chloroform protocol. We sequenced the genome on a PacBio RS II instrument using the P4-C2 chemistry. Additional PacBio sequencing data of the same YPS128 genotype were incorporated from and older assembly (*Yue et al., 2017*). A total of 9 SMRT cells were used to produce 1352628 reads, corresponding to approximately 205x genome coverage. We ran the de novo assembly using the hierarchical assembly protocol RS_HGAP_Assembly3.3 with an expected genome size of 12 Mb. Data were deposited at Sequencing Read Archive (SRA), accession number PRJNA622836.

### Very long read (Oxford nanopore) sequencing

To exclude confounding effects of very early mtDNA changes, i.e. during freezing, thawing and the first round of paraquat cultivation, we thawed and single streaked frozen cells from founder (A7 position), A7 $t_{50}$ and D1 $t_{50}$ populations. We isolated and expanded one clone from each population and cultivated these in the presence of paraquat until stationary phase. DNA was extracted using Qiagen Genomic-tip 100 /G DNA extraction kit. Libraries for Oxford Nanopore sequencing were prepared using 1D Native barcoding genomic DNA with the EXP-NBD104 and SQK-LSK108kit. The flow cell version was FLO-MIN106, and the raw nanopore reads were basecalled by guppy (v2.1.3) with a minimal quality score cutoff of 5 (options: --qscore_filtering --min_qscore 5). For all basecalled reads that passed the quality filter, demultiplexing was further performed by guppy with the help of the guppy_reads_classifier.pl from LRSDAY (v1.3.1) (*Yue and Liti, 2018*). The de-multiplexed reads were processed by LRSDAY (v1.3.1) for adapter trimming, reads down sampling (down sampled to 50x coverage), de novo assembly, assembly polishing, assembly scaffolding, and dotplot visualisation. We deposited data at Sequencing Read Archive (SRA), accession number PRJNA622836.

### Resequencing of adapted populations and populations released from selection

We thawed and subsampled frozen chronological record populations and cultivated cells in liquid medium in presence of paraquat overnight (24 hr). DNA was extracted using a modified protocol of the Epicentre MasterPure Yeast DNA Purification Kit. Pool sequencing was performed at SciLife (Stockholm, Sweden), using Illumina HiSeq2500, 2 × 126 bp. Libraries were prepared using the Nextera XT kit to accommodate the low DNA yield from small cultures. At least two founder controls were included in each flow cell.

### Calling de novo point mutations

Sequenced reads were quality-trimmed and nextera transposase sequences were removed with TrimGalore (v.0.3.8). Reads were mapped to the YPS128 pacbio assembly (see above) using BWA MEM (v.0.7.7-r441). PCR and optical duplicates were flagged using Picard-tools (v.1.109 [1716]). Base alignment quality scores were calculated using samtools calmd (v.0.1.18 [r982:295]) and variants were called using Freebayes (v0.9.14–8-g1618f7e). All alleles were reported regardless of frequency or genotype model. Variants were annotated using SnpEFF (v.3.6c). Variants below a quality score of 20 and variants present in the sequenced founder samples were filtered out. Data were deposited at Sequencing Read Archive (SRA), accession number PRJNA622836.

### Calling aneuploidies

Aneuploidies were called using a sliding, non-overlapping 200 bp window coverage of reads mapped. Reads with a MAPQ of <1 were not counted. The window coverage ratio was calculated as $\log_2(k_w/w_{founder, i})$, where $w_i$ is the depth of coverage of mapped reads in each 200 bp window,

$w_{founder}$ is the depth of coverage of each $i$ in a founder sequenced in the same flow cell, and $k = \sum_{i=1}^{G} D_{founder} / \sum_{i=1}^{G} D_{sample}$, where $G$ is the YPS128 genome size and $D$ is the depth of coverage for each nucleotide. Aneuploidies were called by determining the median $\log_2$ window coverage for each chromosome.

## Calling mtDNA copy number change

mtDNA copy number was calculated for each sample using a sliding, non-overlapping window of 1 kB. The mtDNA copy number relative to the euploid nuclear genome was calculated for each window as: $log_2(W_i/W_{median,\ euploid})$, where $W_{median,\ euploid}$ is the median of all 1 kB windows of the nuclear genome, excluding chromosomes with detected aneuploidies. We estimated the median absolute number of mtDNA molecules across all windows, assuming one copy of the nuclear genome and no sequencing bias for mitochondrial DNA, as $2^{\mathrm{median}(\log_2(W_i/W_{median,\ eupoloid}))}$.

# Numerical model of evolving cell populations

## Cell population parameters

To generate simulated adaptation trajectories based on empirical effect sizes and mutation rates of point mutations and aneuploidies, we used an individual-based model implemented in Python (*Gjuvsland et al., 2016*). We repeated each simulation 1152x. We started from a haploid, isogenic founder population that was subsampled at the end of each growth cycle to found the next cultivation cycle. The population parameters were population size at the start of each growth cycle ($N$), the number of cell divisions before subsampling in each growth cycle ($M_t$) and the total number of growth cycles (n=50 cycles). When the total population size reached $2^{Mt}N$ cells, $N$ cells were sub-sampled randomly to found the next cycle. $N$ was set to equal the approximate mean across all empirical sub-samplings. $M_t$ was set to equal the mean (across populations) empirical measure in each growth cycle $t$. Each cell divided 12x before it died. Mating, meiosis, sporulation or ploidy change were not included, and there was no population structure.

## Mutation effect sizes

We estimated the mutation effect sizes empirically. To estimate gene loss-of-function mutation effect sizes underlying simulations shown in *Figure 1D*, we used the haploid BY4741 single gene deletion collection (*MAT**a**;his3Δ1;leu2Δ0;met15Δ0;ura3Δ0; genex::kanMX*) (*Giaever et al., 2002*), as above. To estimate chromosome duplication effect sizes, we used the duplications of chromosome II, III, V, X, and XVI constructed by backcrossing, as above. We reconstructed the chromosome duplications IV, VI, VIII, IX, XI, XII, XIII, XIV, and XV as in *Zebrowski and Kaback, 2008*. We genetically modified the founder clone genotype to match the *his3Δ* (complete deletion by transformation with pSH47) and *can1::STE2pr-HIS3* genotype of the aneuploidic construct, as described in *Zebrowski and Kaback, 2008*, and used these as controls. Duplications of I and VII could not be obtained by either methods, despite repeated tries. Strains carrying duplications were cultivated in absence and presence of each stressor (n=9) and doubling times, $D$, extracted.

## Mutation rate parameters

All cells began as identical, haploid founder cells. Cells had 4947 nuclear encoded protein genes, and 16 chromosomes specified by the sequenced reference genome (R64-1-1). Cells had no mitochondrial genome. Essential genes were not included. Cells independently and randomly acquired nuclear genome mutations as chromosome duplications and point mutations in protein coding genes at the end of each cell division. Mutation rates were constant, equal for all genomes, for all chromosomes and for all nucleotide sites. Chromosomes and nucleotide sites were only allowed to mutate once. Sites on new chromosomes did not mutate. Chromosome duplications occurred at rate of $\mu$=4.85*10$^{-5}$ duplications/cell division. Point mutations occurred at a rate $\mu$=0.33*10$^{-9}$ point mutations/bp/division (*Zhu et al., 2014*).

## Mutation effect size parameters

We tracked the mutations of each cell, its reproductive age, and its cell division time. Mutations and cell division time were passed to daughter cells. Cells began at a cell division time equal to the

founder population doubling time. Change in cell division time was affected by mutations only, and mutations only affected cell division time. Because chromosome duplication and loss-of-gene function point mutations are the most common drivers of adaptive evolution in experiments (*Chevereau et al., 2015*), we assumed these to be the only sources of change in cell division time. We estimated the cell division effect size of chromosome duplications as described above. We estimated the cell division effect size of point mutations by downloading the SIFT yeast database (http://sift-db.bii.a-star. edu.sg/public/Saccharomyces_cerevisiae/EF4.74/) and extracting all possible stop gain base changes and nonsynonymous mutations, with attached SIFT scores (*Vaser et al., 2016*). All stop gain base changes and all nonsynonymous mutations with a SIFT score <0.05 affected cell division time with an effect size equal to the population doubling time effect of the corresponding gene deletion. Reproductive age did not affect cell division time and there were no cell-cell interactions. We assumed that the cell doubling time under stress at any instance could not become shorter than the measured mean founder cell doubling time in absence of stress. We implemented the well documented principle of diminishing return of mutations with increasing fitness (*Chou et al., 2011*; *Khan et al., 2011*), by letting a mutation $m$ define the cell division time, $D_m$, obtained from the equation, $D_m = k(D_G - D_{founder, nostress}) + D_{founder, nostress}$ where $D_G$ is the cell division time of the genotype before the mutation occurred, and $k = max((2^{D_r}D_{founder, stress} - D_{founder, nostress})/(D_{founder, stress} - D_{founder, nostress}), 0)$. $D_{founder, stress}$ is the measured mean doubling time of the founder in presence of stress, and $D_r$ is the estimated doubling time effect size of the mutation assuming no epistasis. No other form of epistasis was included.

## Quantitative PCR of mtDNA genes in evolving cell populations

To track the copy number dynamics of mtDNA genes in evolving populations, we performed quantitative PCR (qPCR) on the frozen chronological samples from a subset of populations (A4, A7, A8, A9, B5, B8, B12, D1). The samples were revived in 3 mL liquid background media supplemented with 400 μg/mL paraquat, expanded to stationary phase (72 hr). DNA was extracted from harvested cells using a MasterPure Yeast DNA purification kit (Epicentre), as per the manufacturer's instructions. Primers were designed for each of the protein and rRNA encoding mtDNA genes and for one nuclear control: *CDC5*. The small size and extreme AT richness of the RNA subunit of RNase P (*RPM1*) prevented the design of working PCR primers for this mtDNA gene. We ran qPCR for duplicates of the entire frozen chronological record of one mtDNA gene in a single run, together with *CDC5* controls. The qPCR was performed using iTaq Universal SYBR Green Supermix (total volume: 20 μL) and run on a Bio-Rad CFX Connect using Bio-Rad Hard-Shell PCR 96-well thin-wall plates sealed with adhesive transparent film. The PCR protocol was: initial denaturation (95 C; 15 min) followed by 45 cycles of: denaturation (95 C; 15 s), anneal (60 C; 30 s), extension (72 C; 30 s), and a melting curve analysis. We quantified PCR products at the annealing step of each cycle due to the low melting temperature of all PCR products, which follows from extremely low GC% of the mtDNA. The relative copy number was calculated as $log_2(2^{-(Ct_{mtDNA} - Ct_{CDC5}) - (Ct_{mtDNA} - Ct_{CDC5})_0})$.

We capped all Ct values at 30. We also set Ct values to 30 for rare (5.6%) sample replicates where the *COX1* primer pair produced non-PCR based background signals.

## Light and fluorescence microscopy of evolving cells

We performed light and fluorescence microscopy on DNA (DAPI) stained $\rho^+$ (founder, WT), $\rho^-$ (population A7 at $t_{50}$), $\rho^-$ (population B8 at $t_{50}$) and $\rho^0$ (*mip1Δ*; lacking the mitochondrial DNA polymerase) cells to validate that $\rho^-$ cells contained mitochondrial DNA. We cultivated cell populations overnight in liquid background medium, diluted pre-cultures in fresh media to OD$_{600}$=0.3 and incubated with agitation in 15 mL growth tubes until OD$_{600}$=0.6. Cells were pelleted by centrifugation at 3,000 G for 1.5 min. The supernatant was discarded and cells were suspended in 1 mL of ice cold 70% ethanol. Cells were fixated by incubation in RT for 5 min. The cells were then pelleted again by centrifugation, washed in non-ionic water and re-suspended in PBS (Fisher Bioreagents BP2944-100) solution. DNA was stained with DAPI (D3571, Merck) at 50 ng/mL just before imaging. Images were acquired using a Zeiss Axio Observer Z1 Inverted microscope with Plan-Apochromat 100 x/1.40 Oil DIC M27 objective and AxioCam MR R3 camera.

## Electron microscopy of evolving cells

We assayed mitochondrial dynamics by electron microscopy before paraquat stress, after 5 hr (~1 cell doubling) of paraquat stress, during long-term paraquat stress (A7, $t_{50}$) and 5 hr after release from long-term paraquat stress (A7, $t_{50}$). Frozen stocks were revived by transfer to solid background medium (with or without paraquat) and cells were cultivated for 5 hr (1 population doubling). Cells were harvested by rinsing the plate with 5 mL liquid background media (with or without paraquat), suspended by stirring with a plastic spreader and pelleted by centrifugation at 600 G for 1.5 min. Pelleted cells were frozen under high pressure using a Wohlwend Compact 03 (M. Wohlwend GmbH, Sennwald, Switzerland). Freeze substitution was performed in a Leica EM AFS2 (Leica Microsystems, Vienna, Austria) by incubating the cells with 2% uranyl acetate dissolved in 10% methanol and 90% acetone for 1 hr at –90 °C (*Hawes et al., 2007*). Freeze substituted cells were washed (2 x) in 100% acetone and the temperature was raised 2.9 C/hr to –50 °C. Cell pellets were broken into smaller pieces to improve resin infiltration. Infiltration of cells was performed using a ladder of Lowicryl HM20 (Polysciences, Warrington, PA) diluted in decreasing acetone concentrations (1:4, 2:3, 1:1, 4:1) followed by three changes in pure Lowicryl. Each step lasted 2 hr. The resin was polymerized with UV light, first for 72 hr at –50 °C and then for 24 hr at room temperature. Resin embedded blocks were sectioned in 70 nm ultra-thin sections using a Reichert-Jung Ultracut E Ultramicrotome (C. Reichert, Vienna, Austria) equipped with an ultra 45° diamond knife (Diatome, Biel, Switzerland). Sections were collected on copper grids coated with 1% formvar and stained with 2% uranyl acetate and Reynold's lead citrate. Stained sections were imaged at 120 kV using a Tecnai T12 microscope (FEI Co., Eindhoven, The Netherlands) and a Ceta CMOS16 camera. The IMOD package (*Kremer et al., 1996*) was used for quantification of the two-dimensional area covered by cells, as a proxy for cell volume, and of the two-dimensional area covered by mitochondria, as a proxy for mitochondrial volume, in 100 cell sections per sample. To validate quantifications, a large subset of images was analyzed by blind-test by a second person. Conflicting quantifications were discarded.

## Confocal microscopy of evolving cells

We also tracked the mitochondrial dynamics by confocal microscopy of fluorescently labelled (Cox4-EGFP) mitochondria in cells before exposure to paraquat, after 7 hr (1 doubling) of paraquat stress, after 79 hr (1 growth cycle +1 doubling) of stress and 7 hr after release from 144 hr (2 growth cycles) of paraquat exposure. We isolated three transformants and ensured that their respiratory growth was normal by spot-assays on glycerol medium. Transformants were cultivated with or without paraquat for 0 hr or 72 hr in liquid medium, the stationary phase cultures were diluted in fresh media with or without paraquat to $OD_{600}$=0.3 and then incubated with agitation until $OD_{600}$=0.6 (exponential phase). The cells were then pelleted by centrifugation at 3000 G for 1.5 min and washed in MQ water, centrifuged and suspended in PBS. Cells were fixated by incubation at room temperature with 3.7% formaldehyde, washed 3 x in PBS, suspended in ProLong Diamond mounting media and directly mounted on slides and imaged in the microscope. Z-stacks of cells were acquired using a Zeiss Axio Observer LSM 700 inverted confocal microscopy with a Plan-Apochromat 63 x/1.40 Oil DIC M27. The signal was averaged between 4 frames to reduce noise. Z-stacks were color-coded according to Z-dimension (slice) using Temporal-Color Code in Fiji/ImageJ v. 1.52. We pre-processed images using the difference of Gaussians (DoG) to enhance the objects to be measured, that is either cells or mitochondria. To measure cells, we calculated the 3D gradient of the DoG filtered images, using triangle algorithm. To measure mitochondria, we calculated the 3D median filter of the DoG filtered images before segmentation. We segmented images using Otsu's method, separating touching objects processed using seed-assisted watershed algorithm. The used seeds were suggested automatically, with a human blinded to sample identities correcting for mistakes. The final quantification of shape descriptors for each cell and the mitochondria within were done in MATLAB, with the relevant code and readme files available at: https://github.com/CamachoDejay/SStenberg_3Dyeast_tools, *Stenberg, 2022* copy archived at swh:1:rev:a047daf337fa05f75f7cb3affb498ed70b6d7703.

## Data and materials availability

Sequence data that support the findings of this study have been deposited in Sequencing Read Archive (SRA) with the accession codes PRJNA622836. The growth phenotyping code can be found at https://github.com/Scan-o-Matic/scanomatic.git; *Zackrisson, 2019*, the simulation code at https://github.

com/HelstVadsom/GenomeAdaptation.git; *Vadsom, 2017* and the imaging code at https://github.com/CamachoDejay/SStenberg_3Dyeast_tools, *Stenberg, 2022* copy archived at swh:1:rev:a047daf-337fa05f75f7cb3affb498ed70b6d7703. The authors declare that all other source data supporting the findings of this study are available at https://data.mendeley.com/datasets/mvx7t7rw2d. All unique strains and stored populations generated in this study are available from the Lead Contact without restriction.

## Acknowledgements

We thank Lars-Göran Ottosson for help and advice with strain construction and design of adaptation experiment, Olga Kourtchenko for help with designing nitrogen-limited environments, and Tom Kirkwood for instrumental comments to an earlier version of this paper. The authors acknowledge support from the National Genomics Infrastructure in Stockholm funded by Science for Life Laboratory, the Knut and Alice Wallenberg Foundation and the Swedish Research Council, and SNIC/Uppsala Multidisciplinary Center for Advanced Computational Science for assistance with massively parallel sequencing and access to the UPPMAX computational infrastructure. The authors acknowledge PacBio sequencing technical support from the Norwegian Sequencing Centre. We acknowledge the Centre for Cellular Imaging at the University of Gothenburg and the National Microscopy Infrastructure (VR-RFI 2016–00968) for assistance with the confocal microscopy.

---

## Additional information

### Funding

| Funder | Grant reference number | Author |
| --- | --- | --- |
| Vetenskapsrådet | 2014-6547 | Jonas Warringer |
| Vetenskapsrådet | 2014-4605 | Jonas Warringer |
| Vetenskapsrådet | 2015-05427 | Mikael Molin |
| Vetenskapsrådet | 2018-03638 | Mikael Molin |
| Vetenskapsrådet | 2018-03453 | Johanna L Höög |
| Cancerfonden | 2017-778 | Mikael Molin |
| Norges Forskningsråd | 178901/V30 | Stig W Omholt |
| Norges Forskningsråd | 222364/F20 | Stig W Omholt |
| Agence Nationale de la Recherche | ANR-11-LABX-0028-01 | Gianni Liti |
| Agence Nationale de la Recherche | ANR-13-BSV6-0006-01 | Gianni Liti |
| Agence Nationale de la Recherche | ANR-15-IDEX-01 | Gianni Liti |
| Agence Nationale de la Recherche | ANR-16-CE12-0019 | Gianni Liti |
| Agence Nationale de la Recherche | ANR-18-CE12-0004 | Gianni Liti |
| Human Frontiers Science Program | LT000182/2019-L | Johan Hallin |

The funders had no role in study design, data collection and interpretation, or the decision to submit the work for publication.

### Author contributions

Simon Stenberg, Resources, Data curation, Software, Formal analysis, Validation, Investigation, Visualization, Methodology, Writing – original draft, Writing – review and editing; Jing Li, Formal analysis,

Visualization, Methodology, Writing – review and editing; Arne B Gjuvsland, Conceptualization, Data curation, Software, Formal analysis, Funding acquisition, Validation, Methodology, Project administration, Writing – review and editing; Karl Persson, Formal analysis, Investigation, Visualization, Writing – review and editing; Erik Demitz-Helin, Carles González Peña, Ciaran Gilchrist, Formal analysis, Investigation, Writing – review and editing; Jia-Xing Yue, Formal analysis, Investigation, Visualization; Timmy Ärengård, Software, Formal analysis, Writing – review and editing; Payam Ghiaci, Formal analysis, Investigation; Lisa Larsson-Berglund, Formal analysis, Validation, Investigation, Methodology; Martin Zackrisson, Resources, Software, Methodology; Silvana Smits, Formal analysis, Investigation, Visualization, Methodology; Johan Hallin, Formal analysis, Funding acquisition, Methodology, Writing – review and editing; Johanna L Höög, Supervision, Funding acquisition, Methodology, Writing – review and editing; Mikael Molin, Conceptualization, Supervision, Funding acquisition, Writing – original draft, Writing – review and editing; Gianni Liti, Conceptualization, Supervision, Funding acquisition, Writing – review and editing; Stig W Omholt, Conceptualization, Supervision, Funding acquisition, Writing – original draft, Project administration, Writing – review and editing; Jonas Warringer, Conceptualization, Supervision, Funding acquisition, Visualization, Methodology, Writing – original draft, Project administration, Writing – review and editing

**Author ORCIDs**
Simon Stenberg ⬤ http://orcid.org/0000-0003-0300-1730
Carles González Peña ⬤ http://orcid.org/0000-0002-7771-7988
Jia-Xing Yue ⬤ http://orcid.org/0000-0002-2122-9221
Johanna L Höög ⬤ http://orcid.org/0000-0003-2162-3816
Mikael Molin ⬤ http://orcid.org/0000-0002-3903-8503
Gianni Liti ⬤ http://orcid.org/0000-0002-2318-0775
Stig W Omholt ⬤ http://orcid.org/0000-0002-8320-4337
Jonas Warringer ⬤ http://orcid.org/0000-0001-6144-2740

**Decision letter and Author response**
Decision letter https://doi.org/10.7554/eLife.76095.sa1
Author response https://doi.org/10.7554/eLife.76095.sa2

## Additional files

**Supplementary files**
• Supplementary file 1. Description of stressor environments used as selection pressures.
• Transparent reporting form
• Source data 1. Primers used for strain construction and qPCR.

**Data availability**

Sequence data that support the findings of this study have been deposited in Sequencing Read Archive (SRA) with the accession codes PRJNA622836. The growth phenotyping code can be found at https://github.com/Scan-o-Matic/scanomatic.git, the simulation code at https://github.com/Helst-Vadsom/GenomeAdaptation.git and the imaging code at https://github.com/CamachoDejay/SStenberg_3Dyeast_tools copy archived at swh:1:rev:a047daf337fa05f75f7cb3affb498ed70b6d7703. The authors declare that all other data supporting the findings of this study are available in the article and at https://doi.org/10.17632/mvx7t7rw2d.1.

The following previously published dataset was used:

| Author(s) | Year | Dataset title | Dataset URL | Database and Identifier |
| --- | --- | --- | --- | --- |
| Warringer J | 2020 | Chronic superoxide distress causes irreversible loss of mtDNA segments | https://www.ncbi.nlm. nih.gov/bioproject/ PRJNA622836 | NCBI BioProject, PRJNA622836 |

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
