## [Editor Report]

Stenberg et al., explore how cells adapt to mitochondrial oxidative stress using yeast as their model system. The authors propose that reversible loss of mtDNA leads to reduced ETC function and diminished free radical production and that this represents an evolved survival mechanism. The idea that reversible loss of mtDNA may be an adaptive response under genetic control that is triggered to permit survival under adverse conditions where oxidative stress is elevated is novel and potentially important, especially given critical questions on how chronic or acute oxidative stress may contributes to loss of mtDNA integrity and mitochondrial dysfunction.

---

## [Decision Letter]

**Decision letter after peer review:**

Thank you for submitting your article "Genetically controlled mtDNA editing prevents ROS damage by arresting oxidative phosphorylation" for consideration by *eLife*. Your article has been reviewed by 2 peer reviewers, and the evaluation has been overseen by a Reviewing Editor and Jessica Tyler as the Senior Editor. The following individual involved in review of your submission has agreed to reveal their identity: Michel B Toledano (Reviewer #1).

Essential revisions:

Comments regarding validity of the proposed mechanism:

1) The reviewers comment that the key conclusions of the paper depend on interpretation of data that is correlative in nature and are concerned that no direct evidence is provided e.g. for the claim that mtDNA loss CAUSES specific adaptation to paraquat. Would you be able to demonstrate that the mtDNA mutation specifically reduces ROS production and that cells with these mutations are therefore able to grow in the presence of PQ.

2) The reviewers point out that yeast mitochondria produce superoxide mainly in the bc1 complex and that loss of Cox1 would be expected to increase electron leak at the bc1 complex. Please address this concern with specific reference to yeast mtDNA mutations and ROS production.

3) To expand on the above concern, an alternative explanation was suggested based on work from Prof. Alexander Tzagoloff, showing that destabilization of mtDNA may lead to irreversible mtDNA loss. The reviewer suggests that under such conditions, a percentage of cells may become rho-minus by amplifying small segments of mtDNA into tandem repeats. This reviewer is concerned that this may be what was detected by qPCR. Consistent with this notion would also be your observation that the copy number of some mtDNA segments is increased above levels seen in the wild type genome.

4) Also, a small fraction of the cells in such a population would be expected to be heteroplasmic. The reviewer suggests that such cells can then rapidly become rho plus by recovering the wild-type mtDNA and thus become homoplasmic and respiratory competent when paraquat is withdrawn. The review suggests that this scenario might result in observations similar to what you report but in this scenario the mtDNA would represent the retained mtDNA from only a few percent of the original cell population and would not have much to do with the majority of cells in the population. We would invite you to specifically address / provide further evidence for why you think that the above scenario (points 3 and 4) cannot explain the data that you obtained.

5) Related to this alternative explanation, One reviewer suggests also carefully reading and discussing some of the early literature of mtDNA genetics in yeast to re-evaluate your claims. Specifically, the reviewer suggests to discuss existing publications showing that increased oxidative stress is a condition for inducing rho- cells in which a specific mtDNA segment is amplified into tandem repeats (see full reviewer comment for details).

6) The authors observed that loss of mtDNA from mip1 cells makes them resistant to paraquat. In fact, loss of mtDNA itself causes a crisis of cell survival followed by adaptation (see Cell. 2009 Jun 26;137(7):1247-58. doi: 10.1016/j.cell.2009.04.014). mip1 cells may simply have reduced paraquat uptake or epigenetic changes to resist superoxide. Thus, mtDNA deletion as a specific adaptive process is unfounded.

7) Given these concerns, one of the reviewers felt that describing paraquat-induced mtDNA mutation as a regulatory "gene editing" program is inappropriate. You may want to address this concern, either by changing the term of arguing why you feel that it is appropriate here.

8) Finally, a fundamental concern is that budding yeast is normally anaerobic and that physiological implication of mtDNA mutations therefore would be quite different from those in mammalian cells. The reviewer suggests that more care should be taken when interpreting your data in terms of their implications to mitophagy, cancer therapy and aging-related diseases in mammals.

Further comments:

9) Figure 5 – loss of rtg2 and rtg3 may affect anaplerosis thereby reducing adaptive growth. It does not necessarily involve an antioxidant mechanism.

10) The data on chromosome duplication (Figure S9) are again just correlative rather than causative to reduced adaptive potential.

11) Figure 2 – Mitochondrial morphology changes in function of culture medium and growth stage. The data are meaningless if no vigorous controls for these parameters are in place.

12) Figure S3B – I am not convinced that the signals correspond to mtDNA. Where are the nuclei in these cells?

13) The discussion contains too many speculations and unfunded claims that are not relevant to the reported data.

14) In the abstract, the authors claim that a regulatory circuitry underlies mtDNA "editing". Where is the "circuitry" that acts on mtDNA?

15) The paper suffers from its style, which is very elliptic, the use of complicated, long sentences, the use of terms such as growth cycles, generations etc… that have not been clearly defined at start.

16) Page 4, and S1 legends says that " we see PQ causes doubling time to increase", but where do we have to see that? It is not clear how the growth rate is calculated? How are experiments performed on solid, liquid medium? The figure only shows gene expression data? What is a growth cycle and what is its length? What do you mean by 240 generations? What is the length of one generation in hours? What is a population, and what is the difference with "clonally reproducing cell populations? At best a picture of the plates used to monitor growth should be shown for one to understand how it is done. How do you calculate that 106 min doubling time reduction equals 49.3 % of the maximum possible reduction? And what is this maximum? Figure 1B is confusing: it is understandable that cells are exposed to PQ enough to adapt, then grown without PQ, and then again with PQ, but over the generations shown in the picture, do one not expect to see adaptation after a few "cycles"?

17) Page 5. How can we compare in parallel mitochondrial morphology and growth if the metrics used in the two experiments are different?

18) Page 6: “cells retained the mtDNA segments that were not lost at near…” revise the grammar of this sentence.

19) Page 7. The experiment described in S6A cannot be used to rule out signaling by H2O2: adding 3 mM H2O2 to cells that have already adapted to PQ, whether or not by use of an H2O2 signal, amounts to a severe H2O2 stress, exacerbated by the lack of a functional respiratory chain (petite cells are more sensitive to H2O2, relative to WT). One way of tackling this question would be to see whether adaptive doses of H2O2 (100-300 microM) prior to exposure to paraquat would speed up growth adaptation or not (cross adaptations have been described in the past). Similarly, the WT PQ adaptative response of cells lacking Yap1 or other antioxidants does not prove anything: signaling by H2O2 is mostly localized in confined areas, and this should persist even in a Yap1 mutant.

20) Page 8. The need of SOD2 for PQ adaptation to occur is not really convincing because of the sickness of SOD mutants in general. Further, it shows that there is no adaptation at 12 microG/mL PQ, but then adaptation occurs at a higher dose, but slower, relative to WT. What is the point authors want to make? That SOD by dismutation of the superoxide anion produces H2O2 needed for signaling ? But, authors already ruled out the need for H2O2 to signal adaptation? Please don’t be too peremptory in your conclusion on this experiment. In addition, it is hard to follow the writer: “in four populations, the copy number…” then “two of these fail to adapt” then “the remaining four populations”, but which ones? Lastly the text of Figure 4c indicates 12.5 mG/mL, but the figure 50?

21) Page 10: “the sustentation of mtDNA deletion” is complicated, rather the occurrence of mtDNA deletions.

22) Page 11. It says “the adaptation to on- or off-target effects of PQ”: clarify. Is the duplication event fixed or reversible?

We invite you to address the above points and to provide stronger support for your key claims – either by providing additional data, further interpreting the current datasets or by reference to the published literature.

*Reviewer #2 (Recommendations for the authors):*

The key data leading to the main conclusion are presented in Figure 3, 4 and S6. mtDNA loss seems to correlate with adaptation and mtDNA loss from the mip1 mutant seems to make cells resistant to paraquat. Figure S6D shows delayed reduction in cell doubling time that coincided with acquisition of a single mtDNA deletion in 4 out of 8 sod2 cultures. Loss of mtDNA in sod2 mip1 double mutant seems to increase adaptation for cell growth. This led to the interpretation that delayed adaptation to superoxide production is due to lower rate of mtDNA deletions. The author concluded that (1) loss of OXPHOS is the predominant mechanism responsible for adaptation; (2) Sod2 plays a role in initiating mtDNA deletion that seems to help adaptation, (3) mtDNA deletion and loss of OXPHOS is under regulatory control by “mtDNA editing”. I have numerous concerns with these conclusions.

(1) These data just show a correlation between mtDNA loss and adaptation. mtDNA is vulnerable to oxidative stress and mtDNA loss after paraquat treatment is not unexpected. No direct evidence is shown to support the idea that mtDNA loss CAUSES specific adaptation to paraquat. The authors observed that loss of mtDNA from mip1 cells makes them resistant to paraquat. In fact, loss of mtDNA itself causes a crisis of cell survival followed by adaptation (see Cell. 2009 Jun 26;137(7):1247-58. Doi: 10.1016/j.cell.2009.04.014). mip1 cells may simply have reduced paraquat uptake or epigenetic changes to resist superoxide. Thus, mtDNA deletion as a specific adaptive process is unfounded.

(2) mtDNA instability and deletions can be caused by many mutations in mtDNA. The use of the term “mtDNA editing” is inappropriate.

(3) Throughout the manuscript, the authors did not consider the dynamics of mtDNA mutations in yeast. Increased oxidative stress is a condition for inducing rho- cells in which a specific mtDNA segment is amplified into tandem repeats. Rho- mtDNAs are typically 1-2 kb in length, and can be rapidly segregated into homoplasmy within 6 generations. As such, the qPCR data for the detection of mtDNA deletions may not reflect the genetic status of most cells in cell populations. The “retained” mtDNA segments may just reflect for formation of rho- mtDNA in a cell population in which most cells are rho-zero. I am surprised that the authors can recover rho- genomes of 6 and 34 kb. Some cells are called rho-is odd. The authors should read the early literature of mtDNA genetics in yeast, please.

---

## [Author Response]

Essential revisions:Comments regarding validity of the proposed mechanism:1) The reviewers comment that the key conclusions of the paper depend on interpretation of data that is correlative in nature and are concerned that no direct evidence is provided e.g. for the claim that mtDNA loss CAUSES specific adaptation to paraquat. Would you be able to demonstrate that the mtDNA mutation specifically reduces ROS production and that cells with these mutations are therefore able to grow in the presence of PQ.

The instability of the molecule makes separating O_2_^•−^ production and breakdown very challenging. Measuring O_2_^•−^ levels would be insufficient to address the causality issue as these would decrease in paraquat adapted cells, even if paraquat adaptation was due to paraquat exclusion or inactivation. However, we maintain that we already have provided compelling evidence for the conclusion that mtDNA loss causes specific adaptation to paraquat.

One of the most efficient ways to test a claim about a causal relation is to do an intervention. Our *mip1*Δ experiment is such an intervention (please see the justification of the representativeness of the *mip1*Δ results in our response to comment 6). Removal of Mip1—the sole mitochondrial DNA polymerase—is the gold standard to validate effects of mtDNA loss/reduction in yeast because in contrast to other proteins associated with mtDNA maintenance, and in contrast to ethidium bromide treatment, Mip1 is not known to affect cellular biology through mechanisms other than those mediated by the mtDNA loss. There is absolutely no support for the possibility that Mip1 would be responsible for paraquat exclusion or inactivation, neither in the literature nor in our data. If the mtDNA deletions had nothing to do with the observed adaptation and were just a side effect due to random oxidative mtDNA damage, then *mip1*Δ cells would not be preadapted to paraquat. But in the original manuscript, we showed that *mip1*Δ cells are indeed preadapted to paraquat.

The *mip1*Δ intervention demonstrates that cells lacking OXPHOS activity are much more capable to grow in the presence of paraquat than wild-type cells. Since the most prominent effect of paraquat is that it causes enhanced O_2_^•−^ production in OXPHOS-active mitochondria by redox-cycling, it follows that the O_2_^•−^ production is lower in *mip1*Δ cells than in wild type cells when they are exposed to the same paraquat concentration. This explains the *mip1*Δ cells’ resistance to paraquat. Indeed, it has been extensively documented that the natural leakage of electrons to oxygen is 2-fold lower in fermenting rho0 yeast cells (*1*). Now, one may object that the total lack of mtDNA in *mip1*Δ cells cannot be directly compared with the more modest mtDNA deletions we observe in the early, but still most prominent, adaptation phase. However, even small deletions in the mtDNA tend to cripple the OXPHOS system (e.g. (*2*)). And our mtDNA segmental deletions cover several protein, rRNA and tRNA encoding genes that are absolutely required for the expression or function of OXPHOS. Moreover, paraquat-mediated shuffling of electrons occurs at complex III, with cytochrome B having the critical enzymatic role (*3*). Cytochrome B is encoded in the mtDNA gene *COB*, and as all our paraquat adapting cell populations lose either *COB* itself, or genes required for *COB* expression, it follows that paraquat dependent O_2_^•−^ production is perturbed in our paraquat adapting cell populations.

As a response to the concerns of the reviewers, we exposed paraquat-adapted populations to menadione and found them to be much more resistant than wild type cells (Figure 3—figure supplement 3B). Like paraquat, menadione enhances mitochondrial O_2_^•−^ production(*4*, *5*). But the compound is structurally completely distinct from paraquat, and if an export or inactivation mechanism was responsible for the adaptation to paraquat, it is unlikely that menadione would be exported or inactivated through exactly the same mechanism. We therefore think it is highly legitimate to claim that the preadaptation of paraquat-adapted cells to menadione is because menadione causes much less ROS production in these cells than in wild-type cells. As the source for the ROS production is the OXPHOS system, it follows that the OXPHOS system has to a large degree become deactivated in cells adapted to paraquat. With reference to the paragraph above, the absolutely most straightforward explanation of this deactivation is that it is caused by mtDNA deletion.

We also added vitamin C to paraquat exposed cells. Addition of vitamin C, an antioxidant that accepts electrons from PQ^+^, the free radical (or ‘damaging’) state of paraquat(*6*), caused the doubling time of not only paraquat exposed wild type cells, but also of paraquat exposed *sod2*Δ cells, to be on par with that of unexposed wild type cells (Figure 1—figure supplement 1C, D). Thus, the paraquat concentration of 400 µg/mL impaired cell growth entirely through its effect on O_2_^•−^ production, and the doubling time adaptation observed must be understood solely as an adaptation to the elevated O_2_^•−^ production.

Finally, we measured the growth of all viable deletion strains in the BY4741 deletion collection in the presence and absence of paraquat. We found a strong tendency for the loss of mitochondrial proteins to lead to paraquat resistance. Moreover, gene deletion strains reported to have low mtDNA copy numbers tended to grow slower than the wild type in absence of paraquat, but better than the wild-type in presence of paraquat (*7*). While these deletion strains are less suitable as controls than *mip1*Δ, because of their many varied functions that affect ROS resistance and growth independently of their effect on mtDNA, the results are still fully in tune with the *mip1*Δ and menadione results, supporting the conclusion that “mtDNA loss CAUSES specific adaptation to paraquat”.

Thus, we find it difficult to acknowledge the possibility that mtDNA deletions leading to shut-down of OXPHOS activity, and consequently shut-down of both natural and paraquat-induced mitochondrial O_2_^•−^ production, is just correlated with the observed paraquat adaptation.

However, the intervention data and the other data discussed above establish only that paraquat causes mtDNA deletion and that this mtDNA deletion causes the adaptation to paraquat. They do not allow any conclusion about whether this is due to unspecific mitochondrial damage or deliberate regulation. We deal with this issue below.

We have renamed the section “mtDNA loss drives the first adaptation phase” to “mtDNA segmental deletions cause the swift adaptation to paraquat”, included new data and analyses, and expanded the text considerably to make our reasoning more explicit.

2) The reviewers point out that yeast mitochondria produce superoxide mainly in the bc1 complex and that loss of Cox1 would be expected to increase electron leak at the bc1 complex. Please address this concern with specific reference to yeast mtDNA mutations and ROS production.

The lost segments cover several genes encoding proteins, rRNA and tRNA. When we talked about the *COX1*-*VAR1* region which was shown in our plots, we were explicitly talking about the genes *COX1*, *COB*, *ATP6*, *ATP8*, *OLI1* and *VAR1*. And due to a lack of well-functioning qPCR probes (the tRNA genes are too short to be targeted by qPCR directly) we were implicitly talking about tRNA genes in between these. The observed loss of multiple mtDNA segments containing genes essential for the expression and function of OXPHOS, and the observed loss of respiratory capacity accompanying mtDNA deletions, hardly open for any other interpretation than what we proposed, i.e. the mtDNA deletions cause a reduced OXPHOS activity and electron flow through the ETC in the affected cells. Without the latter, and, in particular, without the enzymatic cytochrome B function in complex III, paraquat is not capable of producing O_2_^•−^ through redox-cycling.

We cannot exclude that deletions of some individual genes in complex IV, e.g. of *COX1* which codes for the enzymatic function of complex IV, may cause increased O_2_^•−^ production through mechanisms driven by natural leakage of electrons to oxygen, or by paraquat. But such single gene deletions are not present in our material, and cannot be, for the simple reason that they would be totally wiped out by natural selection because of their negative effect on growth rate. The adaptive gain in growth rate during the first adaptation phase is huge, which implies that cells with a reduced growth rate due to increased O_2_^•−^ production would not be able to propagate in a population. In other words, after the first adaptation phase, every cell population produces less O_2_^•−^ than what they did immediately after becoming exposed to paraquat because the vast majority of mitochondrial genomes possess mtDNA deletions that have effectively shut down OXPHOS activity.

In the section “mtDNA segmental deletions cause the swift adaptation to paraquat**”** we now mention tRNAs specifically, and we have included a paragraph under Discussion explaining why we will not find mtDNA deletions causing enhanced O_2_^•−^ production.

3) To expand on the above concern, an alternative explanation was suggested based on work from Prof. Alexander Tzagoloff, showing that destabilization of mtDNA may lead to irreversible mtDNA loss. The reviewer suggests that under such conditions, a percentage of cells may become rho-minus by amplifying small segments of mtDNA into tandem repeats. This reviewer is concerned that this may be what was detected by qPCR. Consistent with this notion would also be your observation that the copy number of some mtDNA segments is increased above levels seen in the wild type genome.

We believe the reviewer envisions a paraquat-induced mtDNA destabilization process that is exclusively driven by unspecific oxidative mtDNA damage and which is accompanied by natural selection for cells with disrupted mtDNAs causing reduced OXPHOS activity. The resulting fragmentation of the mtDNA genome, with cells amplifying different small fragments into tandem repeats, which are retained and inherited through unknown mechanisms, would in this model be what we detect with qPCR and the sequence data. This explanation can in principle account for the intervention data and the other data alluded to above. However, it is strongly contradicted by the sequence data, the qPCR data, the Rtg2/3 data and the data on respiratory ability.

As shown in Figure 6—figure supplement 2A, sequence coverage across both retained and deleted regions, is remarkably even. This is fully consistent with segmental deletions, but highly inconsistent with fragmentation and amplification of different small segments to various extents. We also visually inspected sequence data for five populations (A7, D1, A11, A5 and D9) using the Integrated Sequence Viewer (https://software.broadinstitute.org/software/igv/), which specifically marks tandem amplifications based on changes in the read orientation of read-pairs. We found tandem amplifications of small fragments to be exceedingly rare and similar in number and size to what we observed in these populations before paraquat exposure. Moreover, the qPCR data for one of our large segments was in fact constituted by several very small qPCR probes complementary to small sections of DNA in adjacent genes along the segment. And it is extremely unlikely that several qPCR probes will show the same mtDNA copy number, and the same change in mtDNA copy number unless they are part of the same continuous mtDNA segment. In the same vein, mtDNA fragmentation and amplification of small segments would not be expected to be consistent with the qPCR data either, as it would be highly unlikely that our small qPCR probes would, by coincidence, correspond to the small mtDNA fragments that have been amplified.

To further demonstrate that amplification of many small mtDNA fragments in different cells can be ruled out, we isolated single cells from two paraquat adapted populations, clonally expanded these and sequenced their genomes with long-read nanopore sequencing. With this sequencing technology many of the reads have a length that approaches the length of the retained segments. The read alignment shows that most of the long reads span the entire, or almost the entire segment, i.e. they correspond to one, intact mtDNA molecule and not many small fragments (Figure 6—figure supplement 3). In contrast, despite the dynamic nature of mtDNA, we rarely observed tandem amplifications of smaller mtDNA segments.

As the majority of the yeast mtDNA is normally mostly composed of linear, tandemly duplicated mtDNA molecules, we probed whether the retained long mtDNA segments existed as tandem duplications in very long mtDNA molecules as well. We did this by letting the PCR be directed outwards from segment ends, across a potential breakpoint (Figure 6—figure supplement 2C). The PCR data show that at least in one cell population, a tandem arrangement of the retained segment exist. And the existence of tandem duplications is also supported by the read alignments (Figure 6—figure supplement 3). Although this result is a side-point, we think it is worthwhile to point it out, as it is entirely in line with normal yeast mtDNA biology.

We exposed *rtg2*Δ and *rtg3*Δ cell populations (*n=*16) to paraquat (400 μg/mL) for 80 generations. In both cases, all cell populations failed to adapt (Figure 5A), and their capacity for respiratory growth was virtually unperturbed at the end of the experiment (Figure 5B). Importantly, the growth of the *rtg2*Δ and *rtg3*Δ populations was similar to that of the wild type in the presence, as well as the absence, of paraquat. This implies that the lack of these proteins did not cause reduced adaptive growth by affecting the cellular growth physiology, such as anaplerosis. Considering the paraquat concentration used, i.e. *rtg2*Δ and *rtg3*Δ cells experiencing the same paraquat stress as wild type cells, the lack of adaptation after 80 generations of selection and the retainment of OXPHOS function, the O_2_^•−^ production was arguably, throughout the 80 generations, on par with the production In the wild type immediately after exposure to paraquat. This puts a very restrictive upper bound on the frequency of mtDNA deletions that are caused by unspecific oxidative damage due to the increase in O_2_^•−^ production following paraquat exposure. It follows that unspecific oxidative damage is incapable of explaining the swiftness of the mtDNA deletion process.

Moreover, our data on respiratory ability clearly show that we are not dealing with a small percentage of cells having become rho-minus. The vast majority of cells must experience an almost complete deprivation of OXPHOS ability during the initial very swift adaptation phase in order for these data to make sense.

An increase in copy numbers of segments of the mtDNA are frequently encountered in connection with clonal expansion. However, due to the above, we find it much more likely that the temporary increase we observe in our data, and its disappearance during restoration back to wild-type levels, is part of the homeostatic regulatory response (as pointed out in the Discussion).

We have now expanded the text under the sections “mtDNA segmental deletions cause the swift adaptation to paraquat” and “The mtDNA deletion process requires mito-nuclear communication” to highlight that the above model is not capable of explaining the data. However, despite some effort, we have not been able to identify papers by A. Tzagoloff that should be cited in this connection. The provision of some further coordinates would be highly appreciated.

4) Also, a small fraction of the cells in such a population would be expected to be heteroplasmic. The reviewer suggests that such cells can then rapidly become rho plus by recovering the wild-type mtDNA and thus become homoplasmic and respiratory competent when paraquat is withdrawn. The review suggests that this scenario might result in observations similar to what you report but in this scenario the mtDNA would represent the retained mtDNA from only a few percent of the original cell population and would not have much to do with the majority of cells in the population. We would invite you to specifically address / provide further evidence for why you think that the above scenario (points 3 and 4) cannot explain the data that you obtained.

Concerning the statement: “Also, a small fraction of the cells in such a population would be expected to be heteroplasmic. The reviewer suggests that such cells can then rapidly become rho plus by recovering the wild-type mtDNA and thus become homoplasmic and respiratory competent when paraquat is withdrawn.” This ‘point 4’ scenario does indeed explain parts of the data concerning the recovery after short-term paraquat stress, and it was part of the explanation we proposed in the original manuscript to account for the rapid restoration of wildtype mtDNA copy numbers in populations released from short-term paraquat stress (see end of section “Sustained mtDNA deletion causes.” in the original manuscript). However, because the data tell us that there is no Darwinian selection pressure favoring the removal of mtDNA genomes causing loss of OXPHOS activity, this scenario does not account for the speed at which this restoration occurs, i.e. it does not explain why there is not a sustained replication of non-intact mtDNA genomes after release from paraquat. In fact, according to the current clonal expansion theory, one would expect that mtDNA genomes possessing deletions would outcompete the wild-type mtDNA genomes. But this is contrary to what we observe.

More specifically, the mtDNA genomes possessing deletions causing cessation of OXPHOS activity are faithfully replicated until the cells are released from paraquat. After the release, the heteroplasmic cells “then rapidly become rho plus by recovering the wild-type mtDNA and thus become homoplasmic and respiratory competent”. But in order for this to happen with the speed we observe, either the replication of the mtDNA genomes possessing deletions has to stop or the standing pool of these mtDNA genomes has to be removed by some sort of mitophagy. Both processes may also be simultaneously operative. In any case, it implies that a considerable amount of regulation has to be introduced in order to explain our experimental data. Such a regulatory machinery is well supported by recent publications, and we now expand on this in the discussion.

If the mtDNA deletions are just due to unspecific oxidative damage then, according to the vast literature on the topic, one would expect that while the cells are exposed to paraquat, the mitophagic machinery would be activated to remove these mitochondria. But it is not. And after release from paraquat, one would expect, according to current clonal expansion theory, that mtDNA genomes possessing deletions would outcompete the wildtype mtDNA genomes in individual cells. But they do not. Instead they disappear remarkably fast. Thus, one may claim that the existing literature implies that the combined ‘points 3 and 4’ explanation is internally inconsistent, as well as being in conflict with the experimental data by not acknowledging any sort of regulatory control of the replication and maintenance of non-intact mtDNA genomes as a function of paraquat exposure.

So, if there is no deliberate regulation involved, why do two genes documented to play a key role in mito-nuclear communication, in particular in connection with mtDNA deletions, have such an impact on the adaptation to paraquat? The fragmentation hypothesis suggested under point 3 is not in conflict with the existence of signaling from the mitochondria to the nucleus as a result of mtDNA deletion. But how does it explain that the supposed random mtDNA deletion process due to oxidative damage is hampered in *rtg2*Δ and *rtg3*Δ populations? The Rtg2/3 signaling from the mitochondria to the nucleus activates a whole range of responses dedicated to compensate for the anticipated damage and restore cellular integrity. Thus, a direct prediction by the «point 3 fragmentation» hypothesis is that if there is a cellular response from deletion of Rtg2 or Rtg3, it should be opposite of what we actually observe, i.e. more damage, more fragmentation and more mtDNA deletion. We therefore think that in this case the «point 3 fragmentation» hypothesis is in direct conflict with the data.

In conclusion: Yes, the ‘points 3 and 4’ scenario is capable of explaining some of the experimental data we provided. But we think it is fair to demand that an alternative explanatory scenario is, at least to some degree, able to account for all the data we presented in the original manuscript and not just a subset. In any case, it should not be in conflict with any of the data, unless it is justified that these data can be neglected. However, the ‘points 3 and 4’ scenario is in direct conflict with the Rtg2/3 results, the qPCR and sequence data. And these can hardly be dismissed as irrelevant. In addition, the ‘points 3 and 4’ scenario is incapable of explaining the dynamics of non-intact mtDNA genomes during and after paraquat exposure. Thus, we think it is fair to claim that this scenario does not represent a valid alternative explanatory scheme.

In addition to the revisions mentioned under the previous point, we revised one of the paragraphs under Discussion to make some of the above reasoning clearer, and we contrast our explanatory scenario with the one proposed above, using their mutually exclusive predictions.

5) Related to this alternative explanation, One reviewer suggests also carefully reading and discussing some of the early literature of mtDNA genetics in yeast to re-evaluate your claims. Specifically, the reviewer suggests to discuss existing publications showing that increased oxidative stress is a condition for inducing rho- cells in which a specific mtDNA segment is amplified into tandem repeats (see full reviewer comment for details).

As explicated above, our data do not support the amplification of tandem repeats of small mtDNA fragments. The newly generated long read Nanopore data show that some of the mtDNA molecules in our paraquat adapted populations exist as linear, tandem duplications of the entire mtDNA segment, as discussed below. But such a duplicated linear arrangement is the normal state for yeast mtDNA and is not in any way in conflict with the argumentation in the paper. So, we do not see the need to re-evaluate our claims. But we have revisited the early literature, and we are incapable to see how this literature contradicts the existence of an active mtDNA deletion process. We may have overlooked something, but to find this information it needs to be pointed out to us.

We have now included to two older references:

– Fangman WL, Henly JW, Churchill G, Brewer BJ (1989) Stable maintenance of a 35-base-pair yeast mitochondrial genome. *Mol Cell Biol* 9(5):1917–1921.

– Maleszka R, Skelly PJ, Clark-Walker GD (1991) Rolling circle replication of DNA in yeast mitochondria. *EMBO J* 10(12):3923–9.

6) The authors observed that loss of mtDNA from mip1 cells makes them resistant to paraquat. In fact, loss of mtDNA itself causes a crisis of cell survival followed by adaptation (see Cell. 2009 Jun 26;137(7):1247-58. doi: 10.1016/j.cell.2009.04.014). mip1 cells may simply have reduced paraquat uptake or epigenetic changes to resist superoxide. Thus, mtDNA deletion as a specific adaptive process is unfounded.

We interpret this comment to say that *mip1*Δ cells will undergo dramatic cellular reconfiguration to adapt to the loss of mtDNA even when growing on a rich glucose medium, and that this reconfiguration causes reduced paraquat uptake or epigenetic changes to resist paraquat. We have already dealt with this in our reply to Comment 1 above. But we want to restate that to the best of our knowledge there is absolutely no evidence in the literature supporting that mip1Δ, or mtDNA loss, would be responsible for paraquat exclusion or inactivation. The cited Cell paper does not provide such a backing. Moreover, it describes a mechanism for mtDNA loss in response to replicative aging, an inherently multifactorial process, and any crisis of cell survival would then need to be considered in the context of the other aging factors also accumulating.

Moreover, in terms of cell division time, our *mip1*Δ populations do by no means indicate that they have undergone a crisis of cell survival. Yes, their doubling time on glucose without paraquat is higher than that of the wild type (3.5. vs. 1.5h), so they certainly suffer from the total lack of mtDNA. But this slowing-down of growth is hardly large enough to justify a claim about “a crisis of cell survival”. We constructed and generated this deletion strain in the lab. Considering the short time span (20-30 cell generations) that passed from the isolation of *mip1*Δ clones to the onset of paraquat experiments, it seems quite unlikely that a purely Darwinian process of mutation and selection would be able to create and fix solutions that dramatically change the physiological configuration of *mip1*Δ cells. As mentioned above, paraquat exposure tends to suppress the growth defects of rho0 deletion strains, which instead enjoy a growth advantage compared to the wild type. Thus, the argument that the preadaptation of the *mip1*Δ cells to paraquat is because of effects other than their complete lack of an active ETC system is hard to reconcile with the data, as well as our knowledge of how paraquat acts.

Moreover, our respiratory capacity experiments demonstrate that a large fraction of paraquat adapted populations was totally stripped of OXPHOS activity shortly after the first adaptation phase. This is consistent with the conception that their large mtDNA deletions prevent the expression of a functional OXPHOS. Thus, the cells in these populations do indeed lack an intact ETC system, and this, in turn, impairs both paraquat-induced O_2_^•−^ production at complex III, as well as base-line O_2_^•−^ production. The fact that these cells homeostatically maintain the copy numbers of undeleted mtDNA, while at the same time growing dramatically better than the wild type, implies that the concerns attached to the *mip1*Δ cells do not apply here. Still, their behavior is fully consistent with the behaviors of the *mip1*Δ cells and the rho0 deletion strains. And, still fully in line with the *mip1*Δ results, the growth rate of every single paraquat adapted cell population becomes much less perturbed by the ROS-inducer menadione than that of wild type populations. One can of course claim that this preadaptation of the paraquat-adapted cells to menadione is not caused by their demonstrated lack of OXPHOS activity due to mtDNA deletions, but that each and every one of them developed independently other mechanisms that prevent uptake of menadione. However, as we find absolutely no support for this idea in the sequence data, we cannot see any biological justification for why the most straightforward explanation should be discarded.

With all due respect, we think our responses to points 1, 2, 3, 4 and 6 show very clearly that the statement “mtDNA deletion as a specific adaptive process is unfounded**”** is not justified. Instead, we think our comments on points 1-6 fully justify that there is ample reason for claiming that mtDNA deletion drives the adaptation to supraphysiological mitochondrial oxidative stress. Given how beneficial these mtDNA deletions are in terms of fitness, and how rapidly they emerge and disappear, the most straight-forward conclusion is that the mtDNA deletion represents an evolutionary favored defense mechanism over which the cell has some level of regulatory control. And this is very strongly supported by the results showing that the process depends on Sod2, Rtg2 and Rtg3.

We now deliberately deal with this concern regarding mip1 cells under the section “mtDNA segmental deletions cause the swift adaptation to paraquat”.

7) Given these concerns, one of the reviewers felt that describing paraquat-induced mtDNA mutation as a regulatory "gene editing" program is inappropriate. You may want to address this concern, either by changing the term of arguing why you feel that it is appropriate here.

According to Merriam-Webster Unabridged one of the meanings of the transitive verb edit is “to alter, adapt, or refine especially to bring about conformity to a standard or to suit a particular purpose». As the mtDNA deletions cause cessation of paraquat-induced supraphysiological mitochondrial O_2_^•−^ production and that the deletion process depends on the retrograde pathway, we think we are fully justified to use the term gene-editing in the sense that the cells deliberately alter their mtDNA profile to suit a particular purpose.

Justification of the term is made more explicit in the Discussion, and now the term is introduced for the first time in the Discussion. In line with this, we have also replaced “edit” with “deletion” in the title.

8) Finally, a fundamental concern is that budding yeast is normally anaerobic and that physiological implication of mtDNA mutations therefore would be quite different from those in mammalian cells. The reviewer suggests that more care should be taken when interpreting your data in terms of their implications to mitophagy, cancer therapy and aging-related diseases in mammals.

We agree that the evolutionary distance between yeast and humans should have been given greater consideration. However, we intentionally used a wild yeast strain for our experiments. The perception that yeast is very strongly oriented towards fermentative metabolism derives from experiments on domesticated strains. Recent comparisons of wild to domesticated yeast show that domestication has led to a loss of respiratory growth capacity in domesticated lineages and a gain of fermentative growth capacity (*8*). Extreme orientation towards fermentative growth may thus constitute a relatively recent yeast adaptation to man-made niches with highly concentrated sugar, while a more extensive aerobic respiration, and concomitant higher O_2_^•−^ production, is a natural part of the biology of wild and ancestral yeast lineages. The physiological implications of mtDNA deletion in the context of aerobic respiration is thus not likely to be very different from those of mammalian cells.

It is therefore entirely reasonable to be open for the possibility that the disclosed mtDNA editing system emerged in the shared ancestor of yeast and men, and that it has been retained in both lineages – as is the case for core components of both the antioxidant defense (e.g. Sod1, Sod2), and mitophagy(*9*–*14*).

We have made these points clearer in the text, while at the same time being less conclusive on whether our results extend to mammals.

Further comments:9) Figure 5 – loss of rtg2 and rtg3 may affect anaplerosis thereby reducing adaptive growth. It does not necessarily involve an antioxidant mechanism.

Indeed, *rtg*∆ strains have a glutamate auxotrophic phenotype (*15*, *16*). The RTG pathway is believed to be activated by glutamate starvation, and the activations result in the induction of Cit1, Cit2, Aco1, Idh1 and Idh2 expression. These enzymes catalyze the first three steps from the Krebs cycle, generating glutamate precursors (*17*). However, our experiments were performed on a synthetic complete medium, in which a very substantial excess of glutamate and other amino acids and nucleotides is added externally. This suppresses and vastly reduces the need for glutamate, and for anaplerosis more generally. Moreover, we would expect anaplerosis to be important in both the presence and absence of paraquat, and perhaps more important in absence of paraquat when growth and metabolic needs are generally higher. However, the growth of *rtg2*Δ and *rtg3*Δ remained as for wild-type cells in both the presence and absence of paraquat. Thus, any anaplerotic effect is clearly within bounds the cells can handle well. Moreover, the fact that cells missing Rtg2 and Rtg3 not only fail to adapt to paraquat, but also retain normal respiratory growth, shows that the mtDNA remains functional in these cells. Clearly, if paraquat was just causing unspecific oxidative damage of mtDNA leading to mtDNA deletion, then one would not observe this mtDNA retainment even if the deletion of Rtg2 and Rtg3 just affected anaplerosis and caused reduced growth. So, this explanation can hardly be reconciled with the Rtg2/3 results.

Thus, we think the most parsimonious explanation of the Rtg2/3 results is that these two proteins are critical for the observed rapid emergence of mtDNA deletions by being mediators of two-way mito-nuclear communication being part of the deliberate regulatory response to excess mitochondrial O_2_^•−^ production. We can, at this stage, not discount completely that glutamate deficiency, e.g. by very extensive oxidative damage to glutamate uptake mechanisms, plays a role in activating RTG under ROS exposure. However, whether glutamate deficiency or a pure ROS signal activate the RTG pathway has no bearing on the conclusion that the RTG pathway is necessary for the mtDNA deletions that drive the O_2_^•−^ adaptation. Hence, speculation along these lines is premature in this paper.

We have addressed this concern in the section “The mtDNA deletion process requires mito-nuclear communication”.

10) The data on chromosome duplication (Figure S9) are again just correlative rather than causative to reduced adaptive potential.

If we only had observed the emergence of chromosome duplications in the later stages of adaptation, then this objection would be fully justified. But, as reported in the original manuscript, we validated a causal effect through an intervention experiment. We backcrossed adapted cells with chromosome duplications to unadapted founder cells over three consecutive meiotic generations to generate highly recombined gametes with and without chromosome duplications (i.e. the intervention). As a part of the paraquat tolerance and a minor part of the loss of respiratory growth co-segregated with two of the chromosome duplications across these three meiotic generations, the data are clearly not just correlative.

We cannot completely exclude that undetected mutations contributing to these effects were located on the duplicated chromosomes. However, we detected no genes that were mutated in more than two populations and no mutations that coincided in time with the early paraquat adaptation. We also ensured that the cells used for backcrossing were chosen from populations containing no, or very few, detectable mutations. Due to this, we think the backcrossing experiment counts as a genuine causality test.

We have slightly revised the text in the section “Chromosome duplications explain the second adaptation phase” to make it more transparent that the data are based on an intervention allowing causal inference.

11) Figure 2 – Mitochondrial morphology changes in function of culture medium and growth stage. The data are meaningless if no vigorous controls for these parameters are in place.

We performed the sampling as diligently as possible, making sure to take cells cultivated with and without paraquat in the exponential growth stage, corresponding to 1 to 1.5 population doubling (or 5-7h). We determined the timing of the exponential growth stage by following the growth in real-time of undisturbed populations cultivated in parallel. At the time of sampling, all cell populations were far from entering into stationary phase, which is associated with a mitochondrial morphological shift. Mitochondrial fragmentation due to oxidative stress has been observed several times in various organisms, in mitotic as well as postmitotic cell types. So, the observed fragmentation is fully in line with what was to be expected.

Minor changes of the text in the section “Mitophagy is not responsible for the swift first adaptation phase”.

12) Figure S3B – I am not convinced that the signals correspond to mtDNA. Where are the nuclei in these cells?

The extent to which DAPI stains the mitochondrial and nuclear genomes depends on staining time, as well as the type of fixation (ethanol or formaldehyde) – due to the general preference of DAPI for AT-rich regions, which are enriched in the mtDNA. In the case of fomer Figure S3B, and in contrast to in former Figure S9B (now Figure 6—figure supplement 2B), the experimental design was intended to maximize the differences between mtDNA and nuclear staining. However, we acknowledge that the staining protocol may cause unnecessary ambiguity.

We have removed the former Figure S3B, as it represents a side-point that has little bearing on the central message of the paper.

13) The discussion contains too many speculations and unfunded claims that are not relevant to the reported data.

Without a more specific reference to what is unfunded and what is not relevant to the reported data, we are not able to act on this concern. In fact, in our opinion, the Discussion contained no unfunded claims and a Discussion section should allow authors to place their results in what they conceive as the relevant context, as long as it is backed up with references to the literature.

The Discussion has been somewhat expanded in order to meet some of the specific concerns of the reviewers.

14) In the abstract, the authors claim that a regulatory circuitry underlies mtDNA "editing". Where is the "circuitry" that acts on mtDNA?

The existence of a regulatory circuitry is implicated almost per definition by the Sod2/Rtg2/Rtg3 results showing that there is a two-way mito-nuclear communication / interaction that depends on the retrograde pathway. We note that Rtg2 and Rtg3 have no other known cellular functions besides “regulation”. The results showing that mitophagy is repressed as long as the cells are exposed to paraquat, and that this repression is lost after release from paraquat, must also be part of this regulatory circuitry. So, we think it is justified to allude to the existence of regulation that involves Sod2/Rtg2/Rtg3. Admittedly, however, we have not explored the role of other components of the retrograde pathway, i.e. of Rtg1, Mks1, Bmh1/2, Grr1 and Lst8. And we agree that we do not need to use the term “circuitry” to make the point.

We now avoid the use of the term *circuitry* in the text.

15) The paper suffers from its style, which is very elliptic, the use of complicated, long sentences, the use of terms such as growth cycles, generations etc… that have not been clearly defined at start.

We acknowledge that we should have given more attention to defining key concepts in the text and not just in Methods, and that we in some cases sacrificed clarity for conciseness.

Key concepts are now defined also in text. And in several cases we have expanded and simplified the text in order to be less elliptic.

16) Page 4, and S1 legends says that " we see PQ causes doubling time to increase", but where do we have to see that? It is not clear how the growth rate is calculated?

We believe most readers to be sufficiently familiar with the growth rate concept to be able to digest the results without a mathematical description of its extraction, which is instead given in the Methods.

How are experiments performed on solid, liquid medium? The figure only shows gene expression data?

The growth dose-response of wildtype cells to paraquat was misplaced and shown in former Figure S7. We now display it in Figure1—figure supplement 1A, where it is shown together with the expression data.

What is a growth cycle and what is its length? What do you mean by 240 generations?

We now describe this in the Results which reads:

“We then used a high throughput growth platform (*18*) to observe how 96 asexually reproducing yeast cell populations (colonies) on solid agar medium adapted to the chosen paraquat dose in terms of change in cell doubling time over approximately 50 cycles of growth from lag to stationary phase (Figure1—figure supplement 1E), each cycle lasting about 72 h. Cell numbers in each colony doubled 2.5-6x in each cycle, and over the 50 growth cycles the populations doubled in size ~240 times, on average. Neglecting cell deaths and assuming synchronous cell divisions, this corresponds to ~240 cell generations.***”***

What is the length of one generation in hours? What is a population, and what is the difference with "clonally reproducing cell populations?

We have now replaced “clonally” with “asexually” to highlight that these are yeast cell populations incapable of sex and dividing mitotically. Further on in the manuscript, we simplify by abbreviating the longer phrase as “populations” or “cell populations” and we believe there should be little cause for confusion. Sexual reproduction is only used in one experiment, to backcross adapted clones to wildtypes, and then this is explicitly stated***.***

At best a picture of the plates used to monitor growth should be shown for one to understand how it is done.

We now show a schematic of the workflow in Figure1—figure supplement 1E. Images of colony scans, depicting how colonies are arrayed on plates, is extensively shown in the published methods paper by Zackrisson et al., 2016, which we refer to in many places. Moreover, this type of colony array is now quite standard in yeast genetics, and an intrinsic property of experiments based on the Singer RoToR, or equivalant, robots. Thus, we don’t think it is necessary to show such images again.

How do you calculate that 106 min doubling time reduction equals 49.3 % of the maximum possible reduction? And what is this maximum?

We now describe this with the sentence:

“Assuming the minimum achievable cell doubling time to be that of the wild type before exposure to paraquat (a mean of 93 min), this corresponded to 49.3% of the maximum possible reduction.”

Figure 1B is confusing: it is understandable that cells are exposed to PQ enough to adapt, then grown without PQ, and then again with PQ, but over the generations shown in the picture, do one not expect to see adaptation after a few "cycles"?

We realize that Figure 1B is hard to digest and we now describe the figure, and its interpretation, more exhaustively in both text and figure legends. The former reads:

“We then tested experimentally whether a Darwinian adaptive process driven by selection of new mutations could account for the observed paraquat adaptation in a stress-release experiment. To this end, we exposed new cell populations to paraquat over many consecutive growth cycles. After each growth cycle, a fraction of the adapting cells was placed in a paraquat-free medium for 1 to 10 growth cycles before being exposed to paraquat once more. The rationale being that if the adaptation is due to accumulation of random mutations, loss of the adaptation would progress gradually and take many growth cycles. All 96 cell populations retained their acquired tolerance to paraquat (mean reduction in cell doubling time: 106 min) for only 1-3 growth cycles before abruptly losing it (Figure 1B). When employing the same experimental procedure to 96 cell populations from each of the two other environments to which adaptation was also fast (arsenic and glycine), we found that despite the presence of a much stronger Darwinian counterselection (Figure 1C), these populations lost their acquired adaptations more slowly and gradually (Figure 1B). Thus, while a Darwinian mutation/selection-based adaptive process could potentially explain the data for seven of the eight tested stressors, the paraquat adaptation could hardly be reconciled with such a process.”

In a classical Darwinian scenario, the loss of the paraquat adaptation during growth in absence of paraquat, emerge as selection against nuclear gene variants underlying this paraquat adaptation and for other variants. To generate the observed quite dramatic drop of mean cell doubling time in paraquat, over just a few generations of Darwinian evolution in absence of paraquat, there would need to be a truly massive disadvantage of the gene variants that caused the paraquat adaptation. But, in Figure 1C we show that there is no disadvantage at all – a paraquat adapted population grows as fast in absence of paraquat as it did before its paraquat adaptation. This gives very compelling support for the conclusion that the recovery from paraquat adaptation, and implicitly also the paraquat adaptation itself, is not due to Darwinian selection on nuclear gene variants – but to something else. We show this else to be regulated deletions of the mtDNA.

The text has been revised as shown by the above quotes.

While we originally defined and described these concepts in the Methods, we agree that also covering these aspects in the Results makes reading more convenient for the reader.

17) Page 5. How can we compare in parallel mitochondrial morphology and growth if the metrics used in the two experiments are different?

Electron microscopy was performed exactly as in the growth experiments – on cells cultivated on solid growth medium. The confocal microscopy was done on cells cultivated in liquid. As the results from the electron and the confocal microscopy data are very similar, the liquid/solid cultivation medium distinction appears irrelevant for the conclusions drawn. Moreover, in the confocal microscopy, we estimated mitochondrial volume by stacking confocal micrographs of sectioned cells. However, stacking of electron micrographs into a complete 3-dimensional image of cells, is time consuming and the sample size would therefore be very limited. We consequently used the 2-dimensional information present in single electron micrographs of thin sections to extract measures of the cell area occupied by mitochondria in cells. The use of the 2D area covered by organelles, and mitochondrial area specifically, as proxy for volume is standard in the field (*19*–*21*). The electron microscopy and the confocal microscopy show similar result: that the size of the mitochondrial network does not change substantially but it is re-arranged from a tubular to fragmented organization following paraquat exposure. That two distinct methods, and metrics, support this conclusion increases our confidence in its validity.

We have now described the experiment design better in Methods.

18) Page 6: "cells retained the mtDNA segments that were not lost at near…" revise the grammar of this sentence.

Revised.

19) Page 7. The experiment described in S6A cannot be used to rule out signaling by H2O2: adding 3 mM H2O2 to cells that have already adapted to PQ, whether or not by use of an H2O2 signal, amounts to a severe H2O2 stress, exacerbated by the lack of a functional respiratory chain (petite cells are more sensitive to H2O2, relative to WT). One way of tackling this question would be to see whether adaptive doses of H2O2 (100-300 microM) prior to exposure to paraquat would speed up growth adaptation or not (cross adaptations have been described in the past). Similarly, the WT PQ adaptative response of cells lacking Yap1 or other antioxidants does not prove anything: signaling by H2O2 is mostly localized in confined areas, and this should persist even in a Yap1 mutant.

We acknowledge that we should have been more careful when addressing this topic, and that our data do not allow us to rule out H2O2 signaling.

We have completely revised the section “The deletion of mtDNA segments requires *SOD2*”.

20) Page 8. The need of SOD2 for PQ adaptation to occur is not really convincing because of the sickness of SOD mutants in general. Further, it shows that there is no adaptation at 12 microG/mL PQ, but then adaptation occurs at a higher dose, but slower, relative to WT. What is the point authors want to make? That SOD by dismutation of the superoxide anion produces H2O2 needed for signaling? But, authors already ruled out the need for H2O2 to signal adaptation? Please don't be too peremptory in your conclusion on this experiment. In addition, it is hard to follow the writer: "in four populations, the copy number…" then "two of these fail to adapt" then "the remaining four populations", but which ones? Lastly the text of Figure 4c indicates 12.5 mG/mL, but the figure 50?

Yes, *sod2*Δ showed an expected enhanced sensitivity to paraquat, which forced us to reduce the paraquat concentration to achieve an initial fitness reduction similar to the initial fitness reduction of the wild type. But we had to do exactly the same for *sod1*Δ*,* showing that the two deletions have a similar negative effect on growth rate compared to the wild type. The deletion of *SOD1* had no effect on paraquat adaptation, though, and we think this is the best negative control for a *sod2*Δ specific effect one can achieve. Thus, if one discards our *sod2*Δ results with reference to the sickness of these cells, then one should at the same time explain why there is such a striking contrast with the equally sick *sod1*Δ cells. We are not aware of any data providing such an explanation.

We acknowledge that we should have done a better job when presenting these data. But the point we wanted to make was the following: The reason for why we did not get any adaptive response in *sod2*Δ at 12.5 ug/mL could be that the actual stress level relative to the one the wild type experienced at 400 ug/mL was lower than anticipated – for some unknown reason. We therefore increased the paraquat concentration to 50 ug/mL to check out this possibility. We did not want to go beyond this concentration as we assumed this would cause too high O_2_^•−^ stress compared to what the wild type experienced. In this case, we did get a very much delayed response in four out of eight populations. In all of these four cases we observed a single large mtDNA deletion that clearly caused the adaptive response. It can very well be that increasing the concentration somewhat more would have caused that all eight populations had shown an adaptive response. But given the Sod1 results, we would still be left with the conclusion that Sod2 is indeed crucial for the swift deletion of mtDNA segments. The fact that we do get a very postponed deletion in the *sod2*Δ case, and not at all in the *rtg2*Δ and *rtg3*Δ cases, strongly suggests that the observed deletions are not due to accidental oxidative damage. One possible explanation is that when the O_2_^•−^ concentration in the matrix gets above a certain level, its auto-dismutation rate may become high enough to trigger a H2O2 signal that induces mtDNA deletion. We have not elaborated on this idea in the paper though. In the absence of more conclusive data, we prefer to remain agnostic as to whether Sod2 has a signaling function that is separate from its enzymatic activity or H2O2 accumulating in the mitochondrial matrix serves as a signal.

We have corrected the detected typo in the figure legend. As mentioned, we have completely revised the section “The deletion of mtDNA segments requires *SOD2*”, and this revision was also guided by the very helpful concern raised in this comment.

21) Page 10: "the sustentation of mtDNA deletion" is complicated, rather the occurrence of mtDNA deletions.

The word was carefully chosen as we wanted to emphasize that we refer to the sustained activity of an active cellular process that results in the continuous emergence mtDNA deletions. The term *occurrence* does not capture this effect.

22) Page 11. It says "the adaptation to on- or off-target effects of PQ": clarify.

We acknowledge that it is not necessary to make this distinction in order to convey the main message in the actual paragraph.

We have revised the paragraph such that it does not make this distinction any longer.

Is the duplication event fixed or reversible?

The duplications are fixed in nearly all populations, and we are sorry for not clearly informing about this.

The text now reads:

“However, all but four endpoint populations carried extra copies of chromosome II (n=29), III (n=21) and/or V (n=16) (Figure 6—figure supplement 3A) at fixation or near fixation (mean frequency: 0.97)”

References

1. A. R. Reddi, V. C. Culotta, SOD1 integrates signals from oxygen and glucose to repress respiration. *Cell*. 152, 224–35 (2013).

2. G. A. Fontana, H. L. Gahlon, Mechanisms of replication and repair in mitochondrial DNA deletion formation. *Nucleic Acids Res.* 48, 11244–11258 (2020).

3. P. R. Castello, D. A. Drechsel, M. Patel, Mitochondria are a major source of paraquat-induced reactive oxygen species production in the brain. *J. Biol. Chem.* 282, 14186–93 (2007).

4. F. A. V. Castro, D. Mariani, A. D. Panek, E. C. A. Eleutherio, M. D. Pereira, Cytotoxicity Mechanism of Two Naphthoquinones (Menadione and Plumbagin) in *Saccharomyces cerevisiae*. *PLoS One*. 3, e3999 (2008).

5. G. Loor, J. Kondapalli, J. M. Schriewer, N. S. Chandel, T. L. Vanden Hoek, P. T. Schumacker, Menadione triggers cell death through ROS-dependent mechanisms involving PARP activation without requiring apoptosis. *Free Radic. Biol. Med.* 49, 1925–36 (2010).

6. B. Sendra, S. Panadero, A. Gómez-Hens, Selective Kinetic Determination of Paraquat Using Long-Wavelength Fluorescence Detection. *J. Agric. Food Chem.* 47, 3733–3737 (1999).

7. F. Puddu, M. Herzog, A. Selivanova, S. Wang, J. Zhu, S. Klein-Lavi, M. Gordon, R. Meirman, G. Millan-Zambrano, I. Ayestaran, I. Salguero, R. Sharan, R. Li, M. Kupiec, S. P. Jackson, Genome architecture and stability in the *Saccharomyces cerevisiae* knockout collection. *Nature*. 573, 416–420 (2019).

8. M. De Chiara, B. P. Barré, K. Persson, A. Irizar, C. Vischioni, S. Khaiwal, S. Stenberg, O. C. Amadi, G. Žun, K. Doberšek, C. Taccioli, J. Schacherer, U. Petrovič, J. Warringer, G. Liti, Domestication reprogrammed the budding yeast life cycle. *Nat. Ecol. Evol.* 6, 448–460 (2022).

9. J. Rutter, A. L. Hughes, Power2: The power of yeast genetics applied to the powerhouse of the cell. *Trends Endocrinol. Metab.* 26, 59–68 (2015).

10. M. B. Toledano, A. Delaunay, B. Biteau, D. Spector, D. Azevedo, (2003; http://link.springer.com/10.1007/3-540-45611-2_6), pp. 241–303.

11. G. Farrugia, R. Balzan, Oxidative Stress and Programmed Cell Death in Yeast. *Front. Oncol.* 2 (2012), doi:10.3389/fonc.2012.00064.

12. G. G. Perrone, S.-X. Tan, I. W. Dawes, Reactive oxygen species and yeast apoptosis. *Biochim. Biophys. Acta – Mol. Cell Res.* 1783, 1354–1368 (2008).

13. D. J. Jamieson, Oxidative stress responses of the yeast *Saccharomyces cerevisiae*. *Yeast*. 14, 1511–27 (1998).

14. P. Moradas-Ferreira, V. Costa, Adaptive response of the yeast *Saccharomyces cerevisiae* to reactive oxygen species: defences, damage and death. *Redox Rep.* 5, 277–285 (2000).

15. Y. Jia, B. Rothermel, J. Thornton, R. A. Butow, A basic helix-loop-helix-leucine zipper transcription complex in yeast functions in a signaling pathway from mitochondria to the nucleus. *Mol. Cell. Biol.* 17, 1110–1117 (1997).

16. X. Liao, R. A. Butow, RTG1 and RTG2: Two yeast genes required for a novel path of communication from mitochondria to the nucleus. *Cell*. 72, 61–71 (1993).

17. M. T. McCammon, C. B. Epstein, B. Przybyla-Zawislak, L. McAlister-Henn, R. A. Butow, Global transcription analysis of Krebs tricarboxylic acid cycle mutants reveals an alternating pattern of gene expression and effects on hypoxic and oxidative genes. *Mol. Biol. Cell*. 14, 958–72 (2003).

18. M. Zackrisson, J. Hallin, L.-G. Ottosson, P. Dahl, E. Fernandez-Parada, E. Ländström, L. Fernandez-Ricaud, P. Kaferle, A. Skyman, S. Stenberg, S. Omholt, U. Petrovič, J. Warringer, A. Blomberg, Scan-o-matic: High-Resolution Microbial Phenomics at a Massive Scale. *G3 (Bethesda).* 6, 3003–14 (2016).

19. P. H. Reis-Barbosa, J. J. Carvalho, M. del Sol, C. A. Mandarim-de-Lacerda, Commentary on mitochondrial stereology in transmission electron microscopy. *Int. J. Morphol*. 38, 26–29 (2020).

20. A. V Loud, A quantitative stereological description of the ultrastructure of normal rat liver parenchymal cells. *J. Cell Biol.* 37, 27–46 (1968).

21. A. A. Gerencser, C. Chinopoulos, M. J. Birket, M. Jastroch, C. Vitelli, D. G. Nicholls, M. D. Brand, Quantitative measurement of mitochondrial membrane potential in cultured cells: calcium-induced de- and hyperpolarization of neuronal mitochondria. *J. Physiol.* 590, 2845–71 (2012).